# Smc5/6 functions with Sgs1-Top3-Rmi1 to complete chromosome replication at natural pause sites

Sumedha Agashe [1], Chinnu Rose Joseph[1,5], Teresa Anne Clarisse Reyes[1,5], Demis Menolfi [1,4], Michele Giannattasio [1,2], Anja Waizenegger[1], Barnabas Szakal [1,5] & Dana Branzei [1,3 ✉]

Smc5/6 is essential for genome structural integrity by yet unknown mechanisms. Here we find that Smc5/6 co-localizes with the DNA crossed-strand processing complex Sgs1-Top3-Rmi1 (STR) at genomic regions known as natural pausing sites (NPSs) where it facilitates Top3 retention. Individual depletions of STR subunits and Smc5/6 cause similar accumulation of joint molecules (JMs) composed of reversed forks, double Holliday Junctions and hemi-catenanes, indicative of Smc5/6 regulating Sgs1 and Top3 DNA processing activities. We isolate an intra-allelic suppressor of *smc6-56* proficient in Top3 retention but affected in pathways that act complementarily with Sgs1 and Top3 to resolve JMs arising at replication termination. Upon replication stress, the *smc6-56* suppressor requires STR and Mus81-Mms4 functions for recovery, but not Srs2 and Mph1 helicases that prevent maturation of recombination intermediates. Thus, Smc5/6 functions jointly with Top3 and STR to mediate replication completion and influences the function of other DNA crossed-strand processing enzymes at NPSs.

[1] IFOM, the FIRC Institute of Molecular Oncology, Milan, Italy. [2] Dipartimento di Oncologia ed Emato-Oncologia, Università degli Studi di Milano, Milan, Italy. [3] Istituto di Genetica Molecolare, Consiglio Nazionale delle Ricerche (IGM-CNR), Pavia, Italy. [4]Present address: Institute for Cancer Genetics, Department of Pathology and Cell Biology, College of Physicians & Surgeons, Columbia University, New York, NY, USA. [5]These authors contributed equally: Chinnu Rose Joseph, Teresa Anne Clarisse Reyes, Barnabas Szakal. ✉email: dana.branzei@ifom.eu

Accurate duplication of genetic information is essential to maintain genome stability, which is often disrupted in cancer cells. Several processes are directed at ensuring correct chromosome duplication and at facilitating DNA repair of replication-associated lesions prior to chromosome segregation. Nuclear architecture and DNA topology affect the spatial positioning of origins of replication, influence the stability of replication forks and are linked to the ability of cells to disentangle catenations and other DNA linkages, such as recombination structures arising during chromosome replication. Therefore, factors that regulate DNA topology and chromatin structure, such as DNA topoisomerases and resolvases that disentangle crossed strand DNA structures are critical for genome integrity[1–3]. Complex genomic regions known as natural pausing sites (NPSs) are predisposed to fragility[4–6] and resemble mammalian common and early replication fragile sites[7–9]. Typically, NPSs contain repeat elements, such as tRNA and transposable elements known as Ty retrotransposon structures and are located in genomic regions whose replication is finalized late, such as centromeres (CENs), DNA replication termination regions (TERs), and ribosomal DNA (rDNA).

Smc5/6 is one of the three structural maintenance of chromosomes (SMC) family complexes, along cohesin and condensin, with roles in chromatin structure organization and DNA topology potentially linked to its ability to bind and compact DNA supercoils[1,10–12]. Smc5/6 safeguards the integrity of repetitive chromosome regions[13–15] and facilitates replication through late replicating regions and NPSs[5,16,17]. In mammalian cells, SMC5/6 associates also with early replication fragile sites[8] and acts jointly with Fanconi anaemia proteins FANCD2-I with which it interacts to prevent mitotic instability and to facilitate DNA repair[18]. The SUMO ligase activity associated with the Smc5/6 complex, Nse2/Mms21 (NSMCE2 in mammalian cells), plays roles in regulating Smc5/6 function towards DNA repair and recombination intermediate processing[19–22]. In humans, mutations in SMC5/6–NSMCE2 lead to genetic disorders characterized by increased chromosome fragility and hallmarks of replication stress[23,24].

SMC5/6 acts synergistically with the RecQ helicase complex Sgs1–Top3–Rmi1 (STR), (BLM–TOP3α–RMI–RMI2 in mammalian cells) to support proliferation in yeast and mouse B lymphocytes[16,22]. STR activity is regulated via Smc5/6-mediated SUMOylation upon replication damage and this facilitates recombination intermediate resolution[19,22,25–27]. However, not all Smc5/6 activities involve SUMOylation, as the SUMO ligase activity of Nse2/NSMCE2 is not essential for viability, differently from Smc5/6 components[19,20,22]. Because NPSs represent an endogenous source of replication fork stalling[28] and prolonged pausing at repeat elements induces recombination[16,29–31], we investigated whether Smc5/6 cooperates with known DNA recombination intermediate regulators in DNA transactions at NPSs.

Here we report that Top3 and Rmi1 components of the STR complex implicated both in the disruption of displacement loops (D-loops) and in the dissolution of double Holliday Junction intermediates[32–34] formed in the process of homologous recombination-mediated DNA repair associate with NPSs genome-wide where they co-localize with Smc5/6. Sgs1 also co-localizes with Top3, Rmi1 and Smc5/6 genome-wide and at replication TERs. Smc5/6 is not essential for Top3 binding but facilitates its retention at NPSs. Upon dysfunction/depletion of either STR or Smc5/6, cruciform junctions formed by the intertwining of two DNA strands (hemicatenanes) and recombination-like intermediates with structural features of reversed forks and double Holliday Junctions, accumulate in similar amounts, indicating functional cooperation between Smc5/6 and STR during replication termination. We identify a suppressor of smc6-56 (smc6-56-sup) that restores normal

binding of Top3 genome-wide and at NPSs but becomes defective in other functions that act complementarily with Sgs1 and Top3 to counteract joint molecule accumulation at NPSs. Using a genetic approach, we find that upon replication stress, smc6-56-sup is additive with loss of STR components and the Mus81–Mms4 endonuclease, but not with deletion mutations in Srs2 and Mph1 helicases that can prevent maturation of D-loops into recombination intermediates[32] or suppress recombination[35]. The results thus indicate that Smc5/6 has tight connections with Top3 within the STR complex and may coordinate other DNA recombination intermediate processing factors to mediate DNA replication completion at NPSs.

## Results

**STR is enriched at NPSs and co-localizes with Smc5/6.** To analyse if STR is enriched at NPSs, we first examined the chromosomal locations of Top3 and Rmi1 subunits of the STR complex in unperturbed conditions when cells are arrested with nocodazole in G2/M using Chromatin immunoprecipitation (ChIP) on chip[16]. Notably, we find that endogenously tagged Rmi1-Flag and Top3-Flag chromatin-association profiles are highly similar to each other (Fig. 1a) and are enriched at NPSs such as tRNA genes (Fig. 1b) and centromeres (CENs) (Fig. 1c), compared to the expected random localization. Because Smc5/6 and Rrm3 are reported to associate with NPSs[16,36], we further analysed the genome-wide overlap of Smc5/6 and Rrm3 with Top3 and Rmi1 clusters. We found statistically significant genome-wide overlap between all these factors (Fig. 1a), besides common enrichment at tRNA and CENs (Fig. 1b, c). Moreover, 56 out of 71 (78.9%) previously identified TERs[37], contain both Smc5/6 and STR complexes, with all of the 58 TERs (81.6%) bound by Smc6 containing also Top3 (Fig. 1d). In total, 67 out of 71 (94.36%) TERs were enriched in Top3 and Rmi1 (Fig. 1d).

As Top3 has functions independent of Sgs1 in meiosis[38,39] and more prominent than Sgs1 in disrupting D-loops formed during recombination[40], we addressed if Sgs1 clusters are similar to the ones of Top3. While Sgs1 had generally poorer enrichment on chromatin compared to Top3 and Rmi1, we could perform ChIP-on-chip using 6HA-Sgs1 expressed from the ADH1 promoter. We observed statistically significant overlap of Sgs1 with Top3, Rmi1 and Smc6 genome-wide clusters in G2/M cells (Supplementary Fig. 1a). Although Sgs1 was not enriched at tRNAs (Supplementary Fig. 1b), it still showed enrichment at TERs to the same degree as Smc6 (Supplementary Fig. 1c). Thus, the STR complex is associated with TERs and at least Top3 and Rmi1 subunits are enriched at several classes of NPSs along Smc5/6 and Rrm3.

We further analysed if Top3 recruitment is dependent on a functional Smc5/6 complex. The genome-wide clusters of Top3-Flag remained largely unchanged in smc6 mutants that cause reduced amounts of Smc5/6 in G2/M (i.e. S-smc6[16]), in smc6-P4 that carries the K239R mutation near the ATPase domain of Smc6, and in smc6-56 that has H379R and I421T mutations in the N-terminus coiled-coil domain of Smc6[41] (Supplementary Fig. 1d). Notably, the genome-wide coverage of Top3 was reduced in both smc6-P4 and smc6-56 mutants (Supplementary Fig. 1d) indicating that Smc5/6 positively influences Top3 binding on chromatin.

We next investigated the chromatin binding properties of the Smc5/6 mutant proteins, Smc6-P4 and Smc6-56, finding them less enriched at different NPSs, compared to WT Smc6 (Supplementary Fig. 2a). Notably, quantitative analysis of Top3 chromatin binding revealed that its enrichment at various NPSs is reduced in smc6-56 and smc6-P4 mutants (Fig. 1e). The association of Smc6 and Top3 at NPSs was mirrored at replication forks stalled close to replication origins with high doses of hydroxyurea, HU

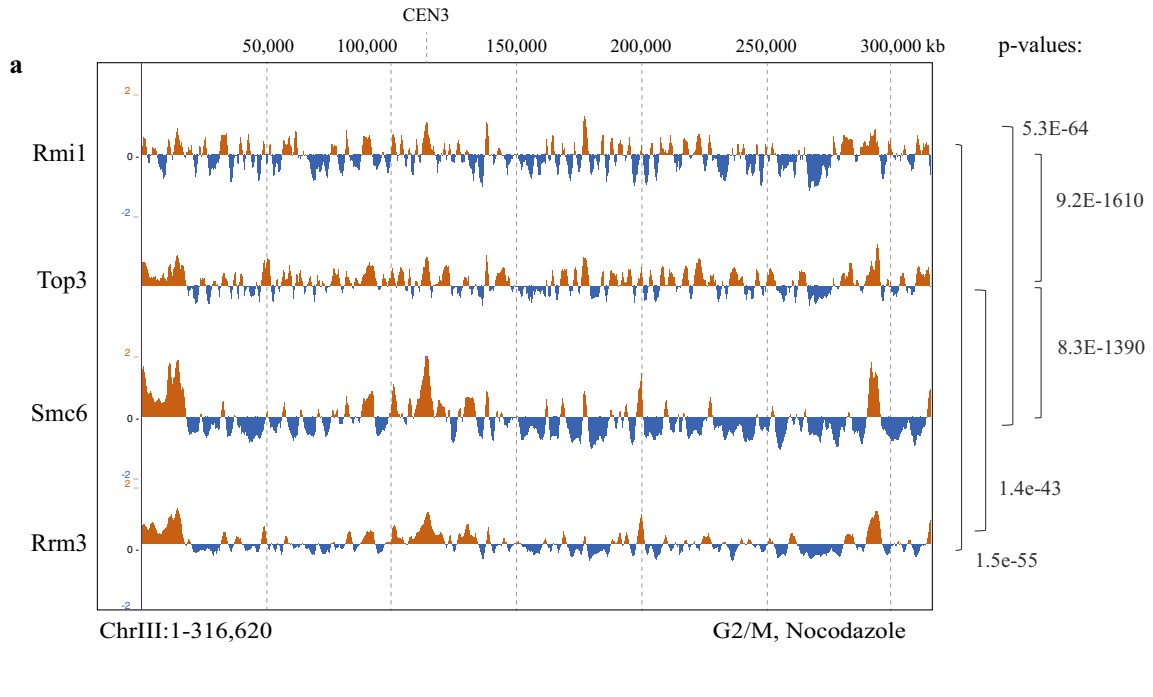

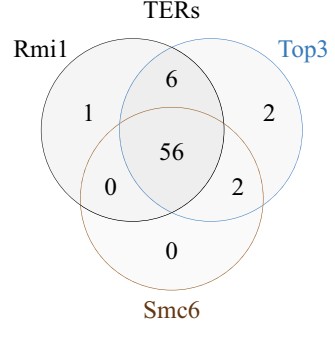

**b**

| tRNA | | |
|---|---|---|
| Protein | Fold increase | p-value |
| Smc6 | 3.03 | 3.6E-1010 |
| Top3 | 2.15 | 1.9E-53 |
| Rmi1 | 2.49 | 1.6E-69 |
| Rrm3 | 2.77 | 2.6E-57 |

**c**

| CENs | | | |
|---|---|---|---|
| Protein | Overlap with CENs | Fold increase | p-value |
| Smc6 | 16 (100%) | 4.07 | 1.4E-12 |
| Top3 | 16 (100%) | 2.60 | 6.6E-08 |
| Rmi1 | 16 (100%) | 3.09 | 2.9E-09 |
| Rrm3 | 16 (100%) | 4.19 | 7.3E-13 |

**e**

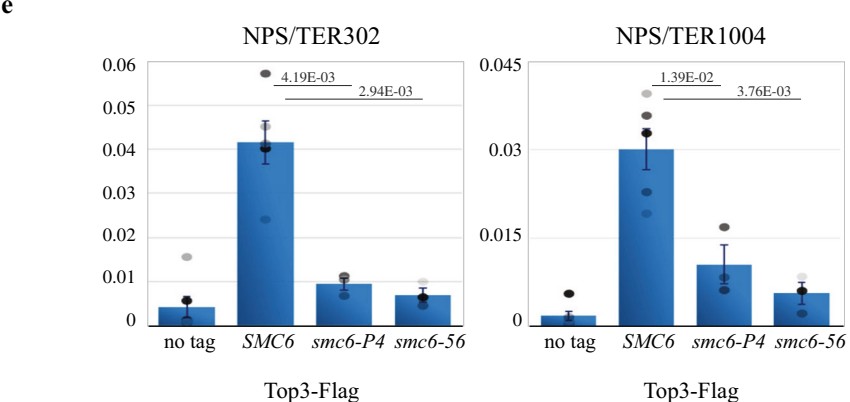

(Supplementary Fig. 2b, c), as previously reported for Smc5/6[16,42]. Thus, Smc5/6 colocalizes with STR genome-wide and facilitates local retention of Top3 at NPSs.

**STR and Smc5/6 prevent joint molecule accumulation**. We next examined the consequences of STR depletion or dysfunction on the replication intermediates arising at NPSs. To this end, we analysed the profile of replication intermediates resolved via neutral–neutral 2D gel electrophoresis (Fig. 2a and Supplementary Fig. 3a) at the late-replicating region, *TER302*[16,37], as well as proximal to the early and efficient origin of replication, *ARS305* (Fig. 2b and Supplementary Fig. 3b). Specifically, we examined the effects of Sgs1 and Top3 depletion using conditional *Tc-sgs1-AID*

**Fig. 1 STR clusters are enriched at NPSs and co-localize with Smc5/6. a** ChIP-on-chip profile of Rmi1-Flag, Top3-Flag, Smc6-Flag and Rrm3-Flag from G2/M-synchronized cells. Chr III is shown as example. The indicated *p*-values (one-tailed Fisher's exact test) relate to the genome-wide overlap between the considered protein clusters. Evaluation of the significance of overlap between the binding clusters of different proteins was performed by confrontation against a null hypothesis model generated with a Montecarlo-like simulation where the "score" for both the randomized positions and the actual data was calculated as the total number of overlapping bases among the whole clusters. The significance of correlation was scored using a one-tailed Fisher's exact test described in detail in the Supplemental Statistical Analysis document in ref. [68]. **b** Analysis of overlap between specific types of elements found at NPSs and clusters of the indicated proteins shows that similar to Smc5/6 and Rrm3, Top3 and Rmi1 are enriched at tRNAs genes. The table reports the fold increase of each protein at tRNA genes, calculated versus the ones expected for random binding, and the *p*-values (one-tailed Fisher's exact test) of significance (see legend of panel **a**). **c** As in **b**, but for centromeres (CENs). **d** Smc5/6, Top3 and Rmi1 are enriched at NPSs that serve as termination sites (TERs) in G2/M. Values of overlap and non-overlap between the protein clusters are shown. **e** ChIP-qPCR profile of Top3-Flag from G2/M-synchronized WT and *smc6-56*, *smc6-P4* cells at different TER regions. The plotted mean value is derived from three biological replicates for *smc6-56* and *smc6-P4* and six biological replicates for no tag and Top3-Flag. Data are presented as mean value ± SEM. The indicated *p*-values were calculated by unpaired two-sided *t*-test. See also Supplementary Figs. 1 and 2. Source data are provided as a Source Data file.

and *Tc-top3-AID* alleles, while keeping the *smc6-56* mutant as control[16]. For these experiments, we used hydroxyurea HU, a condition that causes replication fork stalling as assessed by flow cytometry (see Supplementary Fig. 3c) and induces association of Smc5/6 and Top3 with stalled replication forks (Supplementary Fig. 2b, c).

The termination region *TER302* is replicated passively by converging forks from *ARS306* and *ARS307* early replication origins (Fig. 2b). WT cells show replication forks on the Y-arc and termination signals on the cone and double-Y arc regions when cells are enriched in late S-phase (3 h in HU), with both these intermediates declining in abundance when replication of the region advances (5 h in HU). Differently from WT, mutants in *smc6*, *sgs1* and *top3* accumulate joint molecules (JMs) on the X-arc that persist even at 5 h in HU when the *TER302* region is almost completely replicated as deduced from the reduction in the Y-arc signal (Fig. 2c).

Concerning the replication intermediates forming upon HU treatment proximal to the early origin of replication, *ARS305*, WT cells show bubbles and Y-arc structures (Supplementary Fig. 3a) that diminish in abundance as cells progress in S phase[43]. Distinctly from WT, *smc6-56*, *sgs1* and *top3* mutants accumulate JMs on the X-arc that persist even when the *ARS305* region replication is nearly complete as deduced by flow cytometry analysis of cell cycle progression and the reduced levels of replication intermediates on the bubble and Y-arcs (Supplementary Fig. 3d). To address possible synergy between Smc5/6 and STR, we further analysed the consequences of having simultaneous loss of function in these factors. Double mutants between *smc6-56* and Sgs1 depletion behaved similarly with single mutations in regard to recombination intermediate accumulation (Supplementary Fig. 4). Thus, the STR complex is critical, along Smc5/6, to avert accumulation of JMs at stalled and converging replication forks.

**STR and Smc5/6 jointly prevent cruciform structure accumulation.** To examine if STR and Smc5/6 dysfunctions cause accumulation of similar or distinct types of JMs at stalled replication forks and to identify the nature of such intermediates, we examined their fine structure using in vivo psoralen crosslinking and transmission electron microscopy (TEM)[44]. Because the recombination-related phenotype at NPSs is prominent in late S phase when other X-shaped molecules associated with replication are resolved in control cells (Fig. 2 and Supplementary Fig. 3), we enriched the replication intermediates arising at 5 h of HU treatment from WT and cells depleted for Sgs1, Top3 and Smc5/6. Specifically, we depleted Sgs1 and Top3 from *Tc-sgs1-AID* and *Tc-top3-AID* cells and used a *smc5/6* mutant in which the Mms21/Smc5/6 is restricted to S phase (*S-mms21*), a condition that allows cell viability but impairs Smc5/6 functions at NPSs[16].

Notably, *S-mms21* causes reduced amounts of Mms21 and destabilization of Smc5/6 in G2/M, thereby resembling conditional depletion of Mms21 and Smc5/6 in G2/M.

A large fraction of replication intermediates in WT and all mutants were composed of normal replication forks and bubbles (Fig. 3). There was no evidence of long stretches of single stranded (ss) DNA at the fork branching points (gapped forks) or of hemi-replicated bubbles, thus differentiating the Smc5/6 and STR mutants from *mec1* and *rad53* checkpoint defective cells[45–47]. The ATR/Mec1 checkpoint prevents replication fork collapse by counteracting abnormal fork processing occurring close to replication origins when cells are treated with high doses of HU[45,46,48]. Of interest, *sgs1*, *top3* and *smc6* mutants showed increased levels of reversed forks (5–8% of all intermediates compared to 1% in WT), that is, four-way junctions potentially arising through fork remodelling[49] and of double Holliday Junctions (dHJs) (7–9% of all replication intermediates compared to complete absence in WT) (Fig. 3). We note that these cruciform intermediates also form in mitotic cells replicating in the presence of DNA damage through recombination-mediated gap-filling[33]. Hemicatenanes, four-way junctions with branches of different length occasionally constituted by the connection of two filaments of the same size, were previously visualized via electron microscopy during DNA replication across GAA repeats[50], upon replication inhibition[51] and as final intermediates in recombination-mediated DNA damage tolerance mechanisms[33]. We visualized hemicatenanes in all backgrounds, including WT, and found them increased in the mutants, with Top3 depletion causing the most penetrant phenotype (17%), similar to *S-mms21* (16%) and more severe than *sgs1* (11%) (Fig. 3). The results thus indicate that Top3 can act independently of Sgs1 at final steps of replication through NPSs and in resolving mitotic recombination intermediates involving hemicatenation. Thus, STR and Smc5/6 functionally cooperate in restricting and resolving reversed forks, dHJs and hemicatenanes at NPSs and stalled replication forks.

**smc6 suppressor unveils tight links between Smc5/6 and Top3.** To better understand Smc5/6 functions and potential connections with STR, we performed a temperature-sensitivity suppressor screen for *smc6-56*, which, based on our sequencing, contains two mutations in the first coiled-coil domain, H379R and I421T (Fig. 4a)[41]. We selected eight spontaneous suppressors that showed normal growth at 37 °C (Supplementary Fig. 5a) and back-crossed them with the temperature-sensitive *smc6-56* parent strain. 5 out of the 8 selected natural suppressors showed 2:2 segregation of temperature sensitivity upon tetrad dissection (Supplementary Fig. 5b), indicating their mono-allelic nature. Initial DNA sequencing of the *SMC6* gene in the suppressor clones revealed in all cases an additional amino-acid substitution of Glycine 358 to Cysteine in the coiled coil region close to the

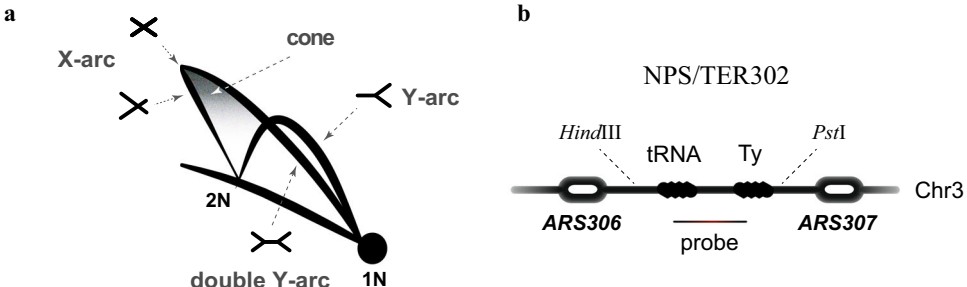

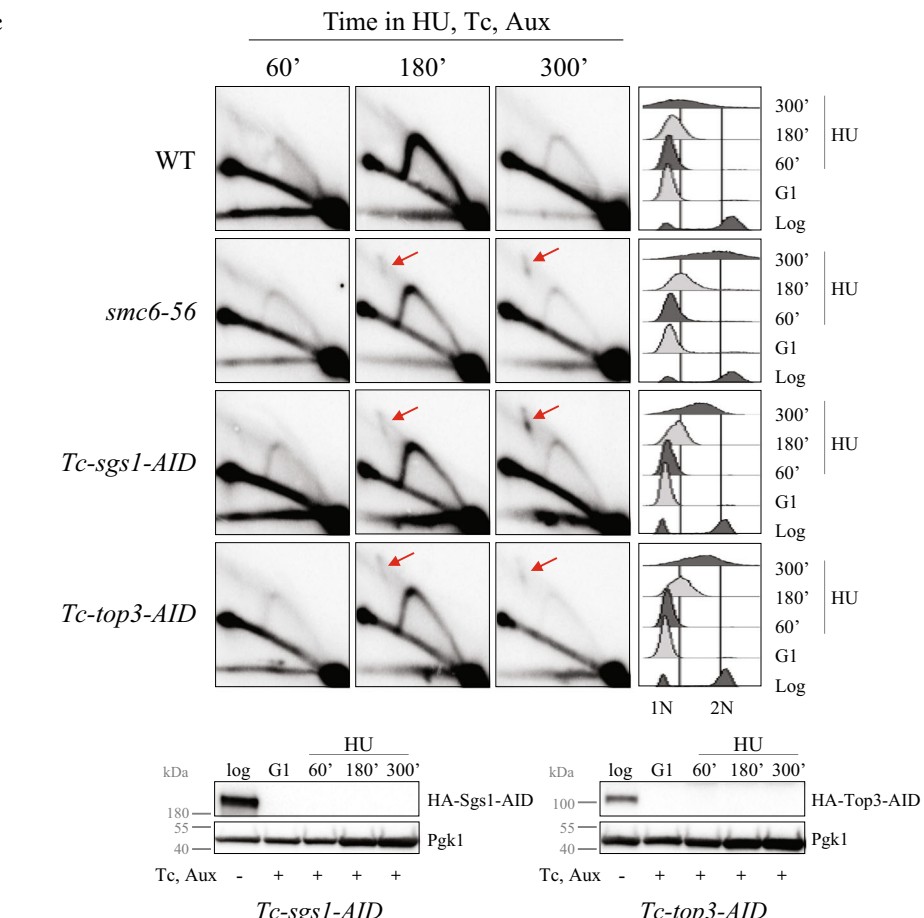

**Fig. 2 STR and Smc5/6 prevent joint molecule accumulation at stalled replication forks and termination regions. a** Schematic representation of replication intermediates arising at termination regions as observed by 2D gel electrophoresis. The migration pattern of different replication intermediates referred to in the text is illustrated. **b** Schematic representation of the *NPS302/TER302* region analysed by 2D gel electrophoresis. **c** Visualization of replication intermediates by 2D gel electrophoresis from cells of the indicated genotype synchronously released from G1 in media containing 200 mM HU, as well as Auxin (Aux) and Tetracycline (Tc). FACS profiles indicated on the right show cell cycle progression. Sgs1 and Top3 tagged with HA were depleted with Auxin and Tetracycline as indicated by Western blots. Pgk1 was used as loading control. Joint molecules accumulating on the X-arc are indicated by red arrows. The experiment was performed reproducibly twice. See also Supplementary Figs. 3 and 4. Source data are provided as a Source Data file.

two point-mutations of *smc6-56* (Fig. 4a). We refer hereafter to the isolated suppressor as *smc6-56-sup*. We recreated this mutation de novo in *smc6-56* and found similar effects with the originally isolated suppressor, demonstrating the suppression capability of this new additional substitution (Fig. 4a). Of interest, the mutations present in *smc6-56* and *smc6-56-sup* map in a domain of the Smc6 coiled-coil neck recently identified as a likely interaction hub for Nse3/4 subunits[11,12]. The isolated *smc6-56-sup* mutant partly stabilized Smc6-56 protein instability at 37 °C (Supplementary Fig. 5c) and fully compensated for the HU and

MMS sensitivity of *smc6-56* (Fig. 4a). Moreover, the synthetic lethality of *smc6-56* with *RRM3* and *SGS1* deletions was suppressed by *smc6-56-sup* (Supplementary Fig. 5d, e).

Notably, when we examined Smc6-56-sup enrichment at different NPSs differing in their features regarding tRNA, Ty content and proximity to CENs, we found reduced chromatin association in comparison with WT Smc6, although higher than Smc6-56 (Fig. 4b). Of interest, the defect in Top3 retention associated with *smc6-56* was fully rescued in *smc6-56-sup* as examined by ChIP-qPCR at different NPSs (Fig. 4c). Moreover,

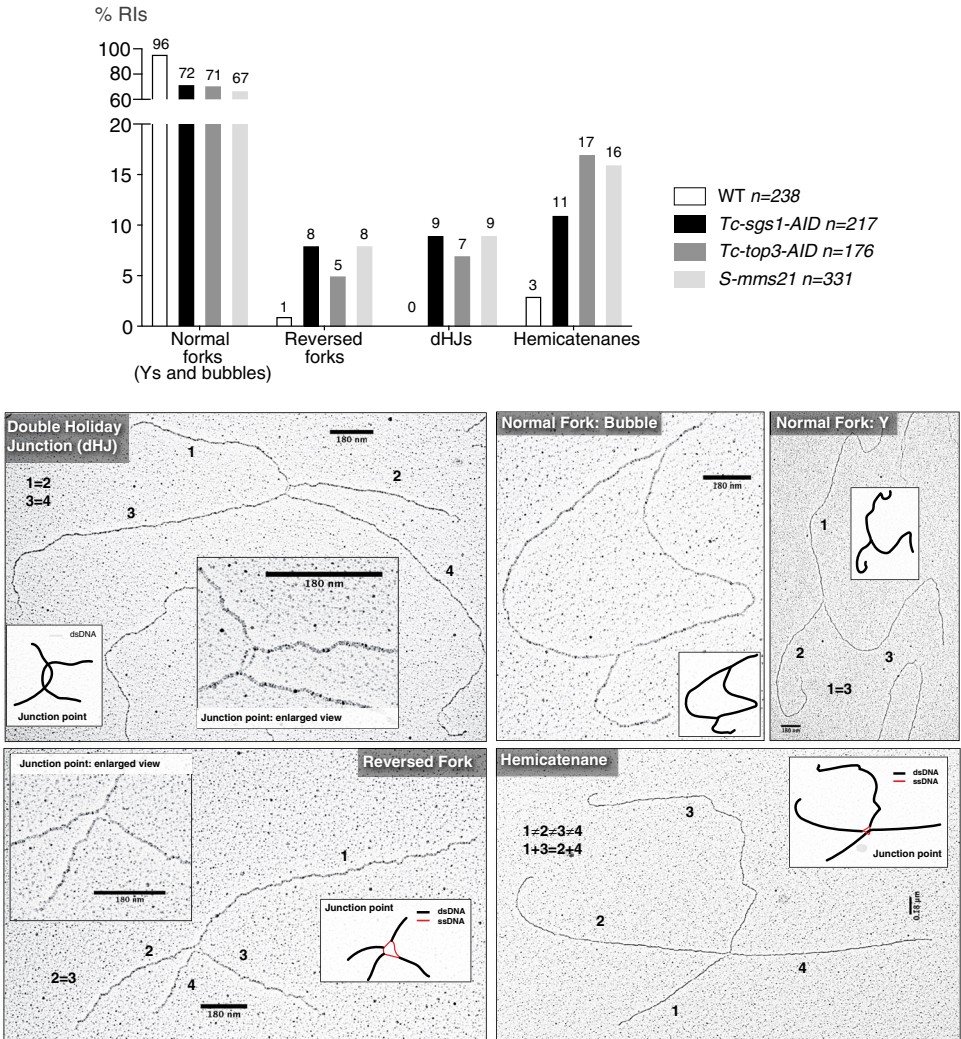

**Fig. 3 STR and Smc5/6 jointly prevent accumulation of hemicatenanes, reversed forks and double Holliday Junctions.** Electron microscopy analysis of replication intermediates and four-way junctions-intermediates in WT and *sgs1*, *top3*, smc5/6 mutants. The plot indicates the percentage of various types of DNA replication intermediates (forks and bubbles) and Joint Molecules (JMs) divided in reversed forks, double Holliday Junctions (dHJs) and hemicatenanes. The number (*n*) of DNA molecules analysed for each genotype is indicated. The intermediates are derived from two biological replicates, with similar results. Typical examples of the visualized categories of JMs are shown with the entire DNA structure and enlarged views of the junction point, with schematic representation of ssDNA regions in red and dsDNA regions in black. The length of the four branches composing the JMs and their relationship in terms of length is shown. Scale bars of 180 nm corresponding to 500 base pairs are indicated. Source data are provided as a Source Data file.

the reduced genome-wide coverage of Top3 in *smc6-56* (see also Supplementary Fig. 1d) was restored in *smc6-56-sup* as observed by ChIP-on-chip analysis of Top3 clusters genome-wide (Fig. 4d). The restoration in Top3 binding in *smc6-56-sup* correlated with a normal profile of replication intermediates, without accumulation of recombination structures (Supplementary Fig. 5f and see below for similar effects at NPSs). Thus, Smc5/6 and Top3 are tightly connected functionally, as the additional Gly358Cys substitution in Smc6-56, although mildly impairing the chromatin binding efficiency of Smc6-56-sup, fully restores Top3 recruitment/retention and relieves the damage sensitivity and synthetic lethality phenotypes associated with *smc6-56*.

**Smc6-56-sup fails to prevent JM accumulation in STR mutants.** Next, we used genetics to test potential defects of *smc6-56-sup* versus the ones of *smc6-56*. Focusing on the roles of Smc5/6 in processing recombination intermediates, we crossed *smc6-56* and *smc6-56-sup* with mutants in different DNA helicases that can destabilize Rad51 filaments and D-loops (*srs2Δ, mph1Δ, sgs1Δ*) or

dissolve dHJs (*sgs1Δ*)[32,35,52], and with mutants in nucleases (*mus81Δ, mms4Δ*) that can resolve single and double HJs to promote chromosome segregation[52–55]. *smc6-56* was synthetic lethal with *srs2Δ* and *sgs1Δ* (Supplementary Fig. 6a and see Supplementary Fig. 5d), but not with *mph1Δ* (Supplementary Fig. 6b) and synthetic sick with *mus81Δ* and *mms4Δ* mutants (Supplementary Fig. 6c). Thus, various DNA recombination intermediate processing factors (Sgs1, Srs2, Mus81–Mms4) are critical for viability in *smc6-56* cells. While *smc6-56-sup* was viable in all tested combinations (Supplementary Fig. 6a–c and see Supplementary Fig. 5d, e), when we examined the HU sensitivity of different mutants, we found that *smc6-56-sup* was additive with *sgs1Δ* (Fig. 5a) as well as with *mus81Δ* and *mms4Δ* (Fig. 5b). These results could be parsimoniously explained by *smc6-56-sup* being a mild hypomorph, with all functions defective in *smc6-56* restored to nearly WT levels but with some defects persisting upon replication stress.

To probe the above hypothesis, we analysed the effect of *smc6-56-sup* in *sgs1Δ* in regard to JM accumulation at TERs. While *smc6-56-sup* behaved like WT (see also Supplementary Fig. 5f),

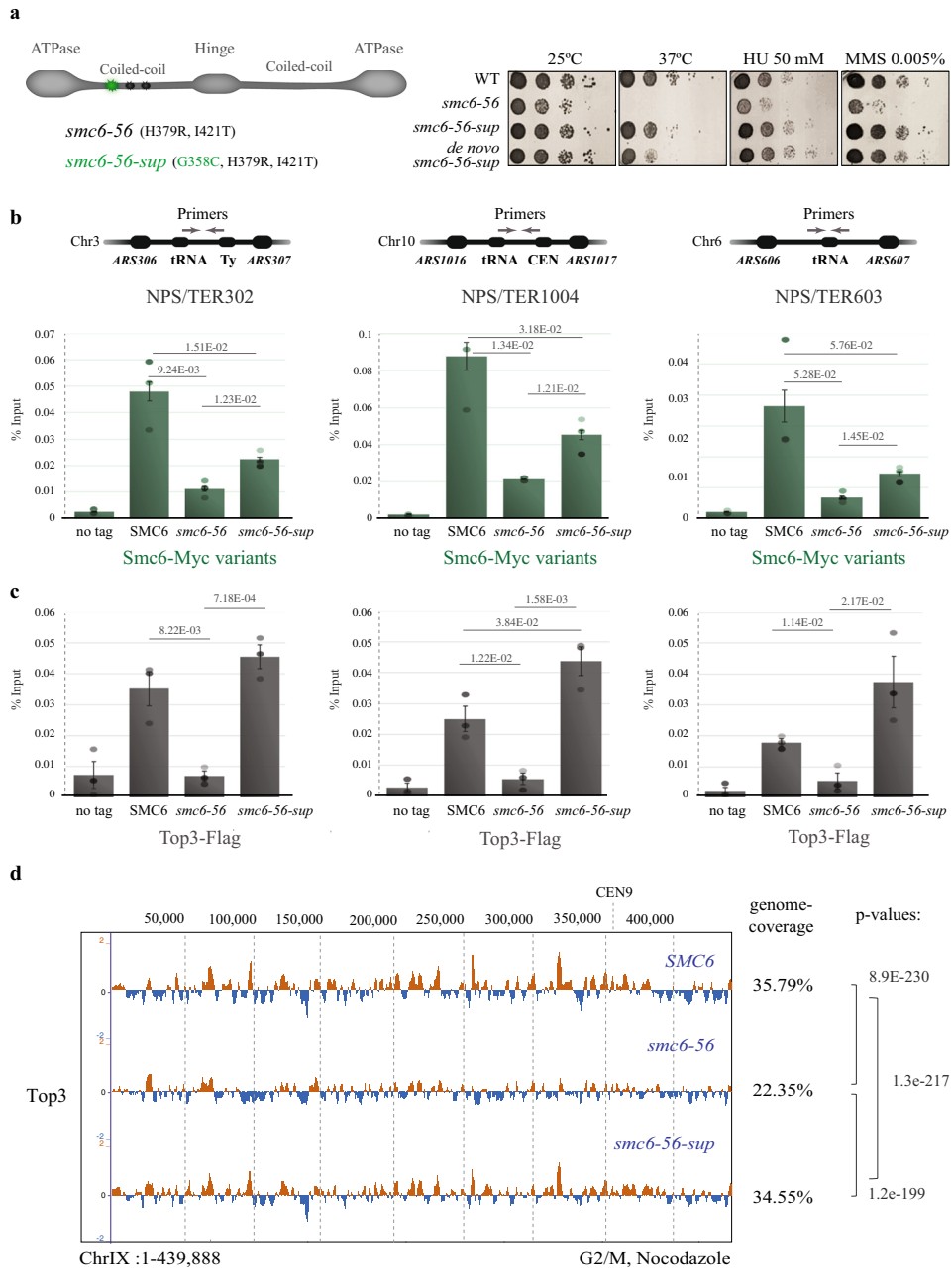

**Fig. 4 Intra-allelic *smc6* suppressor unveils tight links between Smc5/6 and Top3. a** Schematic representation of the *smc6-56* suppressor screen. Representation of the point mutations in *smc6-56*-sup and validation of the suppressor by creating the suppressor mutation de novo in *smc6-56*. **b** ChIP-qPCR analysis of Myc-tagged Smc6 and variants versus no tag in G2/M phase at three indicated NPSs. **c** ChIP-qPCR analysis of Top3-Flag in WT, *smc6-56* and *smc6-56-sup* versus no tag in G2/M phase at three indicated NPSs. For ChIP-qPCR (**b** and **c**), the % Input is derived from three biological replicates and data are presented as mean value ± SEM. *p*-values were calculated by an unpaired two-tail *t*-test. **d** ChIP-on-chip profile of Top3-Flag from G2/M-synchronized WT, *smc6-56* and *smc6-56-sup* mutant cells performed in two biological replicates. Chr IX is shown as example. The indicated *p*-values (one-tailed Fisher's exact test) relate to the genome-wide overlap between the considered protein clusters (see legend of Fig. 1a for details). Genome coverage percentage of Top3 is indicated. See also Supplementary Fig. 5. Source data are provided as a Source Data file.

*smc6-56-sup sgs1Δ* cells showed increased accumulation of JMs at *TER302*, beyond the level observed in *sgs1Δ* mutants (Fig. 5c). Similar synergy in regard to JM accumulation at *TER302* was observed also between *smc6-56-sup* and Sgs1 depletion (Supplementary Fig. 6d). We note that the extended accumulation of JMs was not evident in *smc6-56 Tc-sgs1* (Supplementary Fig. 4). Thus, *smc6-56-sup* may be defective in resolution activities that backup Sgs1, a defect less obvious in *smc6-56* cells. We further analysed whether *smc6-56-sup* aggravates the proliferation defects and the accumulation of JMs associated with Top3 depletion. This was

the case for both phenotypes (Fig. 6a, b). We conclude that *smc6-56-sup* is not only a hypomorph of *smc6-56* but also becomes additionally or more defective in functions that act complementarily with Sgs1 and Top3 to prevent JM accumulation at replication termination regions.

**Smc5/6 coordinates other DNA recombination regulators.** Several types of dysfunctions in *smc6-56-sup* could lead to increased levels of JMs at TERs when STR functionality is impaired. Inability to disengage D-loops, an early intermediate in

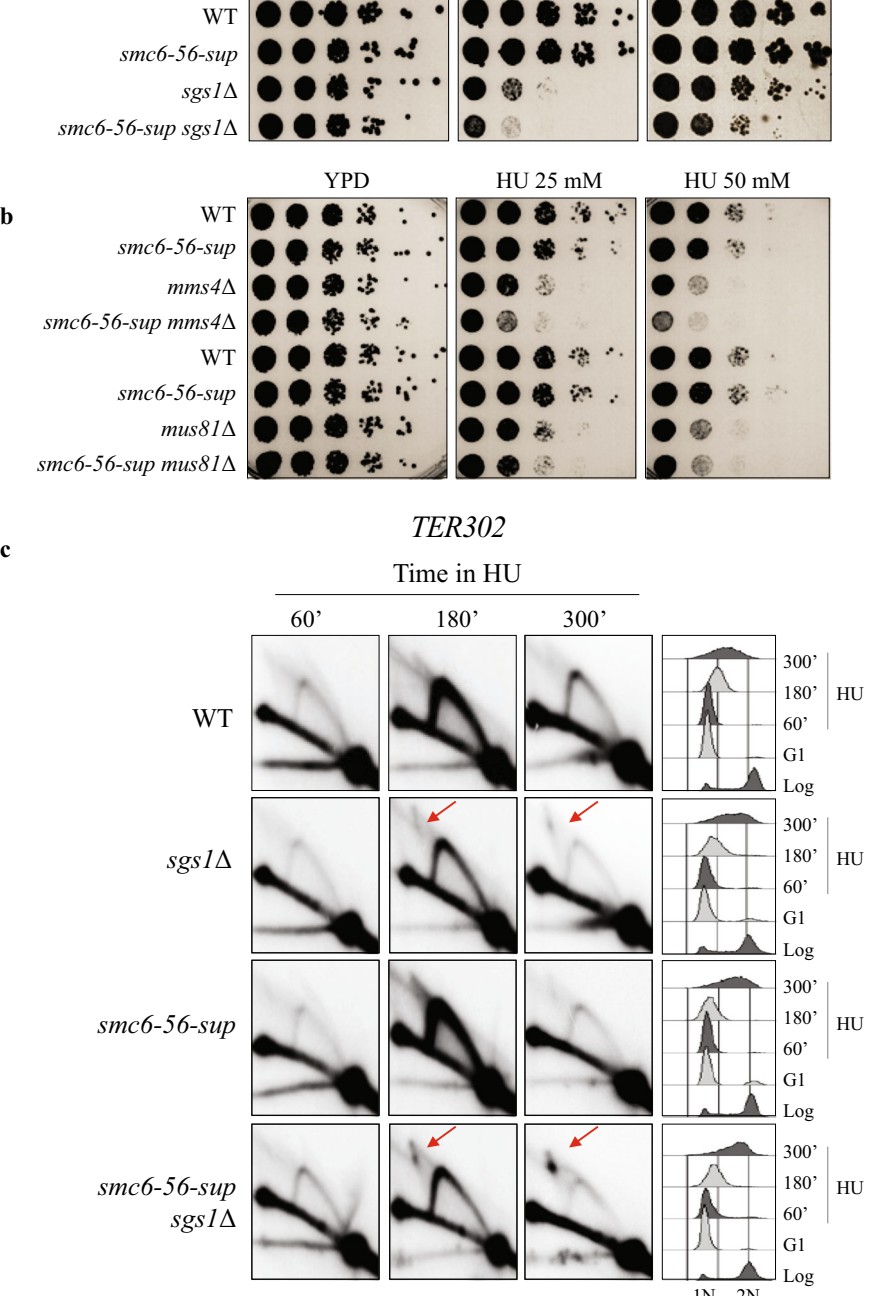

**Fig. 5 Smc6-56-sup protein is defective in pathways complementary with Sgs1 and Mus81. a, b** Additivity between *smc6-56-sup* and *sgs1Δ*, *mus81Δ*, *mms4Δ* in regard to HU sensitivity. The experiments were repeated independently twice with similar results. **c** Visualization of replication intermediates at *TER302* by 2D gel electrophoresis from cells of the indicated genotype synchronously released from G1 in media containing 200 mM HU. The experiment was repeated independently twice with similar results. Flow cytometry profiles are indicated on the right. Joint molecules accumulating on the X-arc are indicated by red arrows. See also Supplementary Fig. 6. Source data are provided as a Source Data file.

recombination, could cause increased levels of JMs. Recently, two main pathways of D-loop resolution activity were defined, one relying on the Srs2 helicase and the other one on the Mph1 helicase and the STR complex[32]. We found that *smc6-56-sup* did not aggravate the sensitivity of either Srs2 or Mph1 mutants (Fig. 7a, b), suggesting potential epistasis or joint function. The lack of additivity with *mph1Δ* may originate from Smc5/6 preventing Mph1-mediated fork reversal, but not D-loop dissociation activity[56], whereas joint roles between Srs2 and Smc5/6 have not been reported yet. A defect in Srs2 activity associated with *smc6-56-sup* could cause more recombination intermediates to

form and may underlie the increased HU sensitivity of Sgs1 and Mus81/Mms4 mutants combined with *smc6-56-sup*, as Srs2 has negative genetic interactions with both STR and Mus81–Mms4 complexes[57]. However, considering that Srs2 loss, alone or in combination with Sgs1 depletion does not lead to a marked increase in JMs upon replication of damaged templates[58,59], the observed accumulation of JMs in *smc6-56-sup sgs1/top3* mutants at NPSs may indicate an additional defect in *smc6-56-sup* associated with resolving mature recombination products.

As the Mus81–Mms4 resolvase is required to resolve persistent JMs accumulating in *sgs1Δ* cells and its activity is increased in

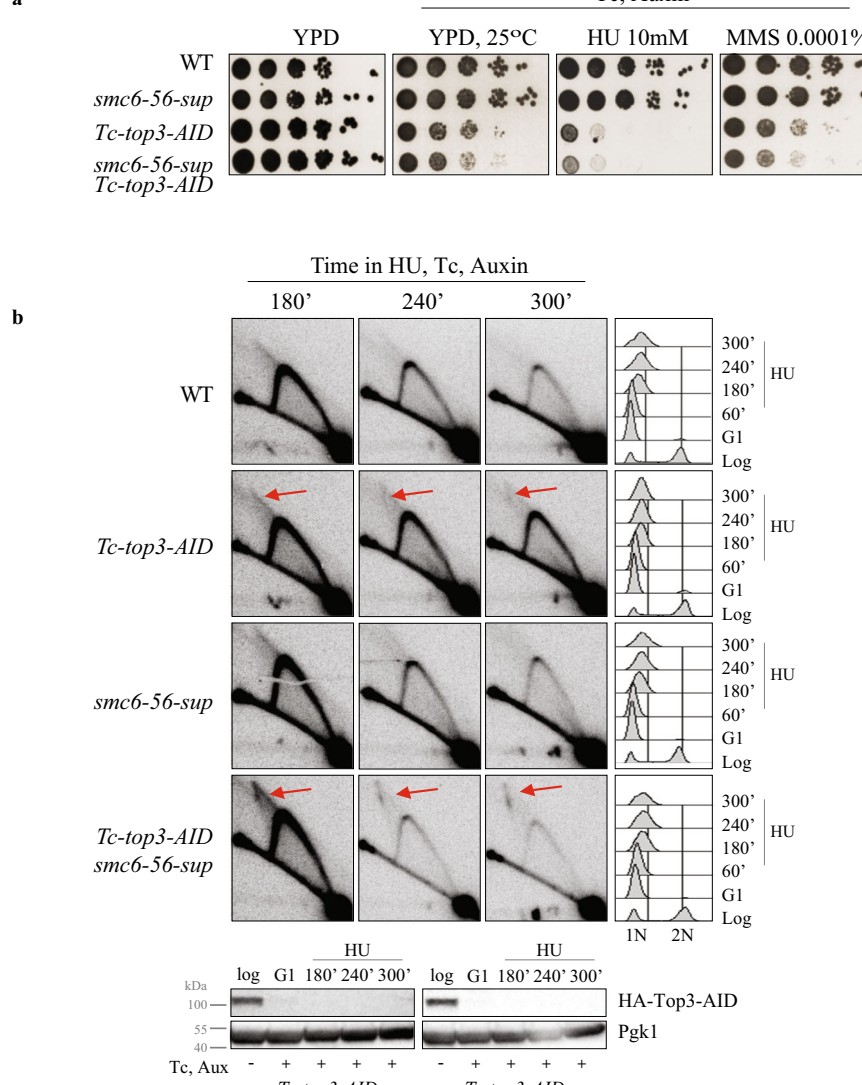

**Fig. 6 Smc6-56-sup is defective in pathways that act complementarily with Top3 to prevent JM accumulation. a** Additivity between *smc6-56-sup* and Top3 depletion induced by tetracycline (Tc) and auxin addition to *Tc-top3-AID*. **b** 2D gel of replication intermediates arising at TER302 in cells released synchronously from G1 in media containing 200 mM HU, tetracycline (Tc) and auxin. The experiment was repeated independently twice with similar results. Flow cytometry profiles are indicated on the right. Joint molecules accumulating on the X-arc are indicated by red arrows. Source data are provided as a Source Data file.

mitosis[54,55,60], one possibility is that the activity of Mus81–Mms4 is also impaired in *smc6-56-sup*. In line with the known cell cycle regulation of Mus81 activity, depletion of Mms4 does not result in JM accumulation at TER regions, differently from *sgs1Δ* (Supplementary Fig. 7a). However, neither *srs2Δ* nor *mms4Δ* caused increased JM accumulation in *Tc-sgs1* cells (Supplementary Fig. 7b), suggesting that the effect of *smc6-56-sup* in increasing the level of JMs accumulating in *sgs1* and *top3* mutants may be due to combinatorial effects on factors regulating JM accumulation and resolution. Of interest, we found that similarly with STR, Mms4 co-localizes with Smc5/6 both genome-wide (Fig. 7c) and at several classes at NPSs, including TERs, CENs and tRNAs, where we find Mms4 significantly enriched (Fig. 7d, e). However, if and how Smc5/6 influences Mus81–Mms4 functionality remains to be clarified, as Mms4 binding at NPSs is not significantly affected by the *smc6-56* and *smc6-56-sup* mutations (Fig. 7f). Altogether, the results indicate roles for Smc5/6 in tuning not only STR but also other factors

with roles in DNA crossed strand processing, potentially including Srs2, Mph1 and Mus81–Mms4, to facilitate disentanglement of catenated and recombination structures arising at NPSs.

## Discussion

Vulnerable regions such as NPSs pose a considerable burden for genome integrity as their replication needs to be assisted by enzymes that facilitate local topology and fork progression[28,36,37]. Although it is now known that NPSs—or features within them, such as tRNA genes, Ty retrotransposons, replication termination sites and early origins of replication—are common features at breakpoints associated with chromosomal rearrangements[4–6], the task-force of factors that mitigates genome instability at these loci and the chromosomal intermediates involved are not well understood[61] (Fig. 8a, b). Smc5/6 is associated with NPSs genome-wide[16,62] potentially due to the propensity of these

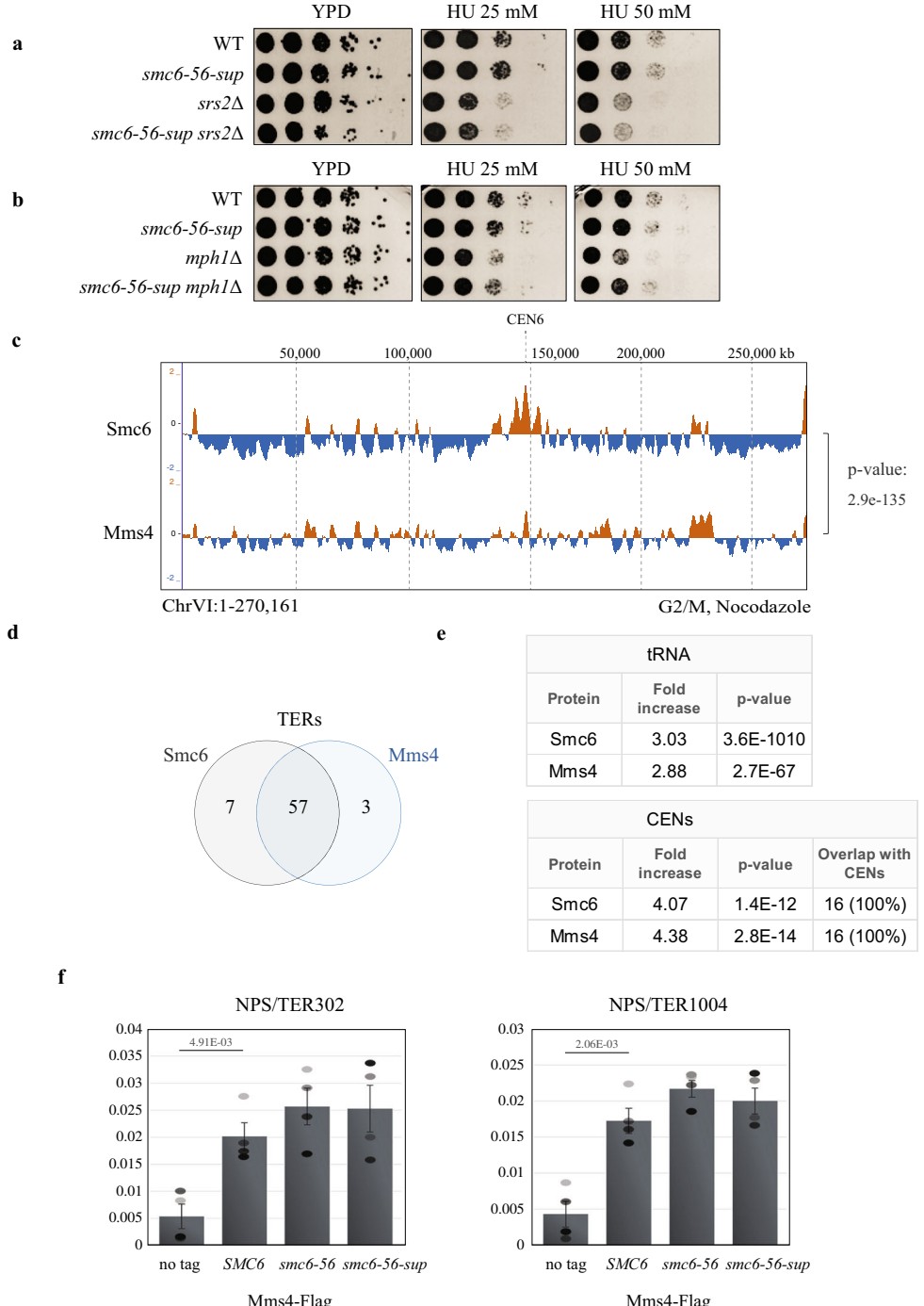

**Fig. 7 Genetic relationship between Smc6-56-sup and DNA processing factors, and Mms4–Mus81 enrichment at NPSs. a, b** No additivity between *smc6-56-sup* and *srs2*Δ, *mph1*Δ in regard to HU sensitivity. The experiments were repeated independently twice with similar results. **c** ChIP-on-chip profile of HA-Mms4 and Smc6-Flag from G2/M-synchronized cells. Chr VI is shown as example. The indicated *p*-values (one-tailed Fisher's exact test) relate to the genome-wide overlap between the considered protein clusters. **d** Mms4 is enriched at NPSs that serve as termination sites (TERs) in G2/M. Values of overlap and non-overlap between Mms4 and Smc6 are shown. **e** fold increase of Mms4 at tRNA genes and centromeres (CENs) calculated versus the ones expected for random binding, and the *p*-values (one-tailed Fisher's exact test) of the significance (see legend of Fig. 1a for details). **f** ChIP-qPCR analysis of Mms4-Flag in WT, *smc6-56* and *smc6-56-sup* versus no tag in G2/M phase at two indicated NPSs. The plotted values are derived from three biological replicates, the data are presented as mean value ± SEM. *p*-values were calculated by an unpaired two-tail *t*-test. See also Supplementary Fig. 7. Source data are provided as a Source Data file.

regions to accumulate DNA supercoiling and intertwines, which can be recognized by Smc5/6 to facilitate local compaction[11,12,63]. Smc5/6 prevents genome fragility in mammalian cells and budding yeast[5,8,18,23,24], but the mechanisms involved remain to be clarified.

Here we provide evidence that Smc5/6 functions together with STR and also other DNA recombination intermediate processing factors in mediating replication termination and integrity of NPSs. We reveal tight functional collaboration between Smc5/6 and STR at NPSs and stalled replication forks where they facilitate

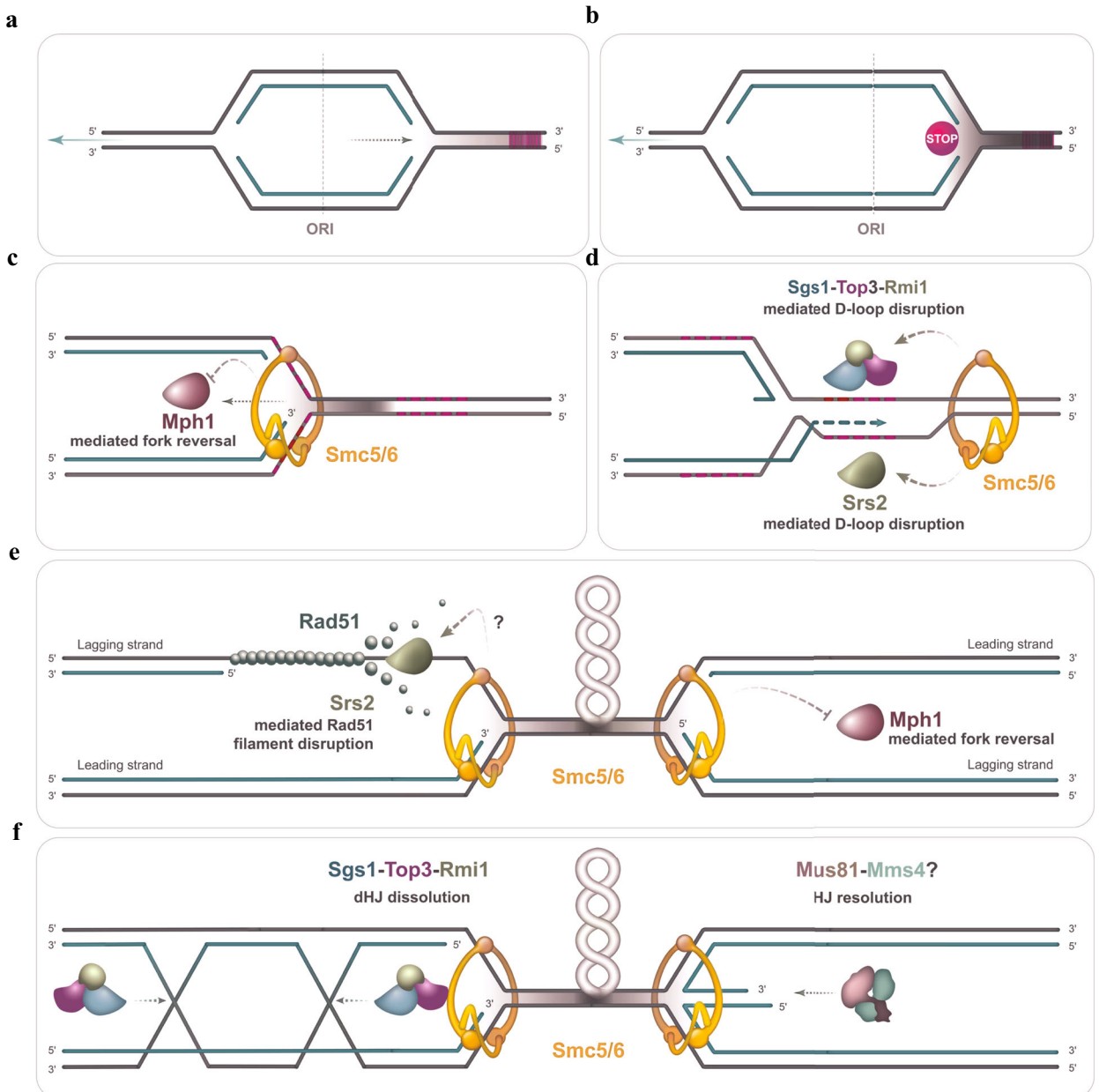

**Fig. 8 Schematic representation of Smc5/6 roles in coordinating STR and other DNA crossed strand processing enzymes upon pausing of replication forks at natural pause sites.** The upper panel shows replication forks progressing normally (**a**) and one fork pausing as it begins traversing elements at a NPS shown in dashed red lines (**b**). Upon fork stalling, Smc5/6 is shown to inhibit the fork regression activity of Mph1 (**c**) and to promote Srs2 and Sgs1–Top3–Rmi1-mediated disruption of displacement loops (D-loops) (**d**). In panel **d**, the D-loops are shown to form by toxic strand invasion, presumably generated upon fork reversal, into repetitive regions (shown as dashed red lines) present in the non-replicated DNA ahead of the stalled fork. As replication forks converge, Smc5/6 may act to manage associated topological stress and prevent formation of recombination intermediates and reversed forks by regulating Srs2 and Mph1, respectively (**e**). At converging forks, in the context of matured recombination structures (**f**), Smc5/6 is shown to facilitate STR complex in processing double Holliday Junctions (dHJs) and Mus81–Mms4 in resolving reversed forks, resembling single Holliday Junctions.

resolution of remodelled replication forks and associated recombination structures, such as transient D-loops and dHJs (Fig. 8). The joint function between Smc5/6 and STR at NPSs is supported by their genome-wide co-localization, Smc5-6-dependent enrichment of Top3, similar types and frequency of recombination intermediates and hemicatenanes that accumulate in the mutants. Importantly, we identify an intra-allelic suppressor of *smc6-56* that restores not only various Smc5/6 functions but also normal Top3 binding. Thus, our findings identify roles for STR at NPSs and highlight that its functionality in

preventing JM accumulation in this context is regulated by Smc5/6 (Fig. 8).

The similar levels of hemicatenanes accumulating in *smc5/6* and *top3* mutants, yet higher than in *sgs1*, indicate that Top3 has roles that are independent of Sgs1, but still coordinated with the ones of Smc5/6. These roles are likely related to the final resolution of hemicatenanes in the step of dHJ dissolution[33,34], as indicated by the EM data, but could also involve D-loop dissociation activity in which Top3 and Rmi1 have more prominent roles than Sgs1 in vitro[40] (Fig. 8d, f). Notably, the similar

percentage of reversed forks and dHJs induced by replication stress in *smc5/6*, *sgs1* and *top3* mutants, pinpoints to functional collaboration between the whole STR complex and Smc5/6 in resolving these structures. Reversed forks, possibly counteracted by Smc5/6 via its action on Mph1[56] (Fig. 8c), could prime toxic recombination within repeats present in the non-replicated region[30,31] (Fig. 8d). The ability of cells to dissociate D-loops or inhibit strand invasion in the first place would be critical to restrict recombination associated with dHJs and final hemi-catenation. Based on genetic evidence, we propose that Smc5/6 does so by regulating not only STR, but also Srs2, which can disrupt Rad51 filaments and D-loops and compensates for the STR complex in recombination intermediate processing[32,57] (Fig. 8d, e).

The model we are proposing draws from our identification of a suppressor mutation of *smc6-56* located in a subunit interaction hub of Smc6[11,12], which unveils defects in pathways that cooperate with Sgs1 and Top3 in preventing JM accumulation at NPSs. Genetically, the defects associated with *smc6-56-sup* do not add up to those caused by individual Srs2 or Mph1 loss, but may involve malfunctions in other DNA resolvases, such as the Mus81–Mms4 endonuclease, which is known to resolve recombination intermediates accumulating in STR[53–55,64] and to act jointly with Mph1 in the repair of DSBs[52]. In potential support of this hypothesis, we find Mms4 to associate with NPSs and co-localize with Smc5/6. Moreover, reversed forks potentially formed by Mph1 could represent good substrates for the Mus81–Mms4 nuclease[65] (Fig. 8e, f). We note that coordination between Smc5/6 and Mus81–Mms4 was also proposed to happen in meiosis[66]. However, as single Srs2 or Mus81–Mms4 loss does not lead to the same effect as *smc6-56-sup* in *sgs1* mutants, we infer that multiple functions become concomitantly impaired and account for the phenotype of the newly isolated *smc6* mutant.

Thus, Smc5/6 may act to both prevent and resolve JMs arising at NPSs (Fig. 8). While these functions are linked to Top3 and STR, they must involve other crossed strand-processing factors, potentially including Srs2, Mph1 and Mus81–Mms4 that act to prevent recombination intermediate formation (Srs2) and maturation (Srs2, Mph1), mediate fork remodelling (Mph1), or resolve single and double HJs (Mus81–Mms4) (Fig. 8). This notion can explain why Smc5/6 is essential for viability, while STR, Mus81–Mms4, Srs2 and Mph1 are not, and provides a basis for understanding Smc5/6 function in the structural integrity of NPSs.

Recent work demonstrated the ability of the yeast and human Smc5–6 complex to locally compact DNA through its ability to bind and stabilize DNA supercoils and intertwines, which are expected to be increased at NPSs and TERs[28,37]. Here we show that at these regions Smc5/6 facilitates the processing of topological and recombination structures by modulating the binding and activity of the Top3 topoisomerase, its associated STR complex and likely of other DNA crossed strand junction regulators (Fig. 8). Thus, Smc5/6 may couple local genome compaction with DNA junction processing, linking replication completion with chromosome segregation.

## Methods

### Yeast Strains.
Chromosomally tagged *Saccharomyces cerevisiae* strains and mutants were constructed by a PCR-based strategy, by genetic crosses and standard techniques[67]. Strains and all genetic manipulations were verified by polymerase chain reaction (PCR), sequencing and phenotype. All yeast strains used in this work are isogenic to W303 background and are listed in Supplementary Table 1.

### Yeast techniques.
Yeast cultures were inoculated from overnight cultures, grown using standard growth conditions and media. All cultures were grown in YPD-media containing glucose (2%) as carbon source at 28 °C, except for the temperature-sensitive strains (*smc6-56* and *top2-4*) that were grown at 25 °C. For

cell cycle synchronization, logarithmic cells grown at 28 °C were arrested in G1 using 3–5 µg/ml of alpha factor for 2–3 h. G2/M arrest was performed with 10–20 µg/ml of nocodazole for 2–3 h. Synchronization was verified microscopically and by flow cytometry (FACS) analysis (see Supplementary Fig. 3c). FACS sample acquisition and analysis was performed using the BD CellQuest Version 3.3. Hydroxyurea (HU) was used at the concentration of 0.2 M unless otherwise indicated. For drug sensitivity and fitness viability assays, cells from overnight cultures were counted and diluted before being spotted in serial dilutions on YPD plates containing the indicated concentrations of HU and incubated at 28 or 30 °C as indicated for 2–3 days.

### Protein-based procedures.
Proteins were analysed from denatured yeast crude extracts. Briefly, $10^8$ cells/ml were harvested, resuspended in 2 ml of TCA 20% and transferred to 2 ml eppendorf tubes. The pellet was resuspended in 200 µl of TCA 20% and an equal volume of acid-washed glass beads (425–600 µm, Sigma) was added. Cells were broken by continuous vortexing for 2–4 min. 400 µl of TCA 5% was added to have a final concentration of TCA 10%. The lysates were then transferred to new 1.5 ml tubes and centrifuged for 10 min at 775×*g* at RT. The pellet was resuspended in 100 µl Laemmli buffer 1×. The pH was then adjusted with 50 µl of Tris Base 1 M. The protein extracts were boiled for 3 min and centrifuged for 10 min at 775×*g* at RT. The supernatant was collected and analysed by SDS–PAGE. Western blot acquisition and quantification was performed using BIORAD Image Lab Version 5.2.1. The following antibodies were used for WB detection: anti-FLAG (mouse monoclonal, clone M2, Sigma; cat. no: F3165, 1:5000 dilution), anti-HA (mouse monoclonal, clone 16B12, Biolegend; cat. no. 901501, 1:3000 dilution), anti-Pgk1 (mouse monoclonal, clone 22C5D8, Novex Life Technologies; cat. no. 459250, 1:5000 dilution), anti-myc (mouse monoclonal), clone 9E10, Santa Cruz Biotechnology cat. no. sc-40, 1:10,000 dilution), anti-mouse IgG HRP-linked (goat, Cell Signaling; cat. no. 7076, 1:20,000 dilution).

### FACS analysis.
For flow cytometry analysis, ~$7 \times 10^6$ cells for each time-point were collected, washed in sterile water, and permeabilized in 70% ethanol at 4 °C overnight. Cells were suspended in 10 mM Tris pH 7.5 buffer, and RNA together with proteins were removed by RNase A (0.4 mg/ml final concentration) and proteinase K (1 mg/ml) treatment. Subsequently, cells were stained with 1 µM Sytox-green (Invitrogen). Cell cycle profiles were obtained following a brief sonication using a Becton Dickinson FACS Calibur system.

### ChIP-on-chip.
For chromatin immunoprecipitation (ChIP), cells were collected at the indicated experimental conditions and crosslinked with 1% formaldehyde for 15–30 min. Cells were washed twice with ice-cold 1× TBS, suspended in lysis buffer (50 mM HEPES–KOH pH 7.5, 140 mM NaCl, 1 mM EDTA, 1% Triton-X100, 0.1% Na-Deoxycholate) supplemented with 1 mM PMSF and 1× EDTA-free complete cocktail, and lysed using FastPrep-24 (MP Biomedicals). Chromatin was sheared to a size of 300–500 bp by sonication. IP reactions, with anti-Flag or anti-Myc antibodies and Dynabeads protein A, were allowed to proceed overnight at 4 °C. After washing and eluting the ChIP fractions from beads, crosslinks were reversed at 65 °C overnight for both SUP and IP. After proteinase K treatment, DNA was extracted twice by phenol/chlorophorm/isoamyl alcohol (25:24:1, v/v). Following precipitation with ethanol and Ribonuclease A (RNase A) treatment, DNA was purified using QIAquick PCR purification kit. For ChIP-on-chip, DNA was then amplified using GenomePlex complete whole genome amplification (WGA) kits WGA2 and WGA3 following manufacturer's instructions. 4 µg of DNA from SUP and IP samples were hybridized to GeneChip *S. cerevisiae* Tiling 1.0R Array (Affymetrix). Primary data analyses were carried out using the Affymetrix Tilling Array Suite Software (TAS), which is used to assess quality of array data. The analysis of clusters was performed as previously described[16,68]. Evaluation of the significance of overlap between the binding clusters of different proteins was performed by confrontation against a null hypothesis model generated with a Montecarlo-like simulation where the "score" for both the randomized positions and the actual data was calculated as the total number of overlapping bases among the whole clusters. The significance of correlation was scored using a one-tailed Fisher's exact test (described in detail in the Supplemental Statistical Analysis document in ref. [68] and the obtained *p*-values are indicated. The antibodies used are anti-FLAG (mouse monoclonal, clone M2, Sigma; cat. no: F3165), 5 µg antibody used for 2.5 x $10^8$ cells; anti-HA (mouse monoclonal, clone F-7, Santa Cruz Biotechnology cat. no: sc-7392), 2 µg antibody used for 2.5 x $10^8$ cells.

### ChIP-qPCR.
ChIP-qPCR was performed using QuantiFast SYBR Green PCR kit according to the manufacturer's instructions and as described in ref. [69]. Each reaction was performed in triplicates using a Roche LightCycler 480 system version 1.1.0.1320. The results were analysed with absolute quantification/2nd derivative maximum and the $2(-\Delta C(t))$ method as previously described[70]. We used primers for TERs *TER302*, *TER603* and *TER1004* previously described in ref. [16] and presented in Supplementary Table 2. Statistical analyses (Student's *t*-test, unpaired, two tailed) were performed with Numbers version 10.3.5 (7029.5.5) and the obtained *p*-values are indicated in figures. All experiments were performed with at least three biological replicates. Antibodies used: anti-FLAG (mouse monoclonal, clone M2, Sigma; cat. no: F3165), 5 µg antibody used for 2.5 x $10^8$ cells; anti-myc

(mouse monoclonal), clone 9E10, Santa Cruz Biotechnology cat. no. sc-40), 5 µg antibody used for $2.5 \times 10^8$ cells.

**2D gel electrophoresis**. Cells were synchronized in G1 phase at 28 °C and released in media containing HU 0.2 M. Samples were collected at the indicated time points and incubated with sodium azide 1% for 30 min on ice. In vivo psoralen cross-linking and DNA extraction with CTAB were performed as in ref. [33] and summarized hereafter. Cells were washed, resuspended in 5 ml of cold water in small Petri dishes and kept on ice. 300 µl of 4,5′,8-trimethylpsoralen solution (0.2 mg/ml in EtOH 100%) was added prior to extensive resuspension by pipetting, followed by 5 min of incubation in the dark and then 10 min of UV irradiation at 365 nm (Stratagene UV Stratalinker 2400). The procedure was repeated three times to ensure extensive crosslinking. Cells were then harvested by centrifugation, washed in cold water, and incubated in spheroblasting buffer (1 M sorbitol, 100 mM EDTA, 0.1% β-mercaptoethanol, and 50 U zymolyase/ml) for 1.5 h at 30 °C. In all, 2 ml water, 200 µl RNase A (10 mg/ml), and 2.5 ml Solution I (2% w/v cetyl-trimethylammonium bromide (CTAB), 1.4 M NaCl, 25 mM EDTA, 100 mM Tris–HCl, pH 7.6) were sequentially added to the spheroblast pellets and samples were incubated for 30 min at 50 °C. 200 µl Proteinase K (20 mg/ml) was then added and the incubation was prolonged at 50 °C for 90 min, and at 30 °C overnight. The sample was then centrifuged at 3200×g for 10 min: the cellular debris pellet was kept for further extraction, while the supernatant was extracted with 2.5 ml chloroform/isoamylalcohol (24:1) and the DNA in the upper phase was precipitated by addition of 2 volumes of Solution II (1% w/v CTAB, 10 mM EDTA, 50 mM Tris–HCl, pH 7.6) and centrifugation at 11,000×g for 10 min. The pellet was resuspended in 2 ml Solution III (1.4 M NaCl, 1 mM EDTA, 10 mM Tris–HCl, pH 7.6). Residual DNA in the cellular debris pellet was also extracted by resuspension in 2 ml Solution III and incubation at 50 °C for 30 min, followed by extraction with 1 ml chloroform/isoamylalcohol (24:1). The upper phase was pooled together with the main DNA prep. Total DNA was then precipitated with 1 volume of isopropanol, washed with 70% ethanol, air-dried, and finally resuspended in 1× TE. Subsequently, 10 µg of DNA were digested with the indicated restriction endonucleases, precipitated with potassium acetate and isopropanol, and resuspended in 10 mM Tris–HCl, pH 8.0. Digested genomic DNA was run on the Thermo Scientific Owl A2 large gel system (gel tray 27 × 20 cm) filled with 2.5 l of 1× TBE. The first-dimension gel (500 ml; 0.35% w/v Agarose D1-LE) was prepared with 1× TBE and run at 50 V for 24 h at RT. The second-dimension gel (500 ml; 0.9% w/v Agarose D1-LE) prepared with 1× TBE was run in the same electrophoresis chamber at 150 V for 12 h at 4 °C with current limited to 150 mA. DNA molecules separated on the second-dimension gels were transferred onto nylon filters via Southern blotting following standard procedures. The DNA samples were digested with HindIII and PstI (for TER302) and NcoI or EcoRV, HindIII (for ARS305) signals were detected following 2D gel electrophoresis and standard southern blot procedures. Signals were detected using probes against ARS305 (BamHI-NcoI 3.0 kb fragment that spans ARS305 and was purified from plasmid A6C-110) or TER302, amplified using primers presented in Supplementary Table 3, and radiolabelled according to the protocol of the Prime-A-Gene labelling system and purified with ProbeQuant G-50 micro columns. The 2D gels were acquired using the Amersham Typhoon Scanner software V1.0 and images were generated with the ImageJ 1.50i software.

**Electron microscopy analysis**. Yeast cells were subjected to in vivo Psoralen DNA inter-strand cross linking and genomic DNA was extracted using the CTAB-Psoralen procedure. The replication intermediates (RIs) were enriched as described previously[44]. Briefly, 15 µg of DNA for each strain were digested with PvuI extensively following manufacturer's instructions and additionally treated with RNAse III to avoid double-strand RNA contamination of the samples (single-strand RNA was removed by RNase A treatment during the genomic DNA extraction). The digestion mix, adjusted to 300 mM NaCl was then loaded on a chromatography column containing 1 ml of BND cellulose stock (0.1 g/column; Sigma B-6385, pre-equilibrated with 10 mM Tris–HCl pH 8, 300 mM NaCl). DNA was incubated with the BND cellulose for 30 min with resuspension every 10 min to allow full binding of the DNA molecules and the flow-through was collected by gravity flow. Twice 1 ml of 10 mM Tris–HCl pH 8 containing 1 M NaCl was added to the column to collect linear double-stranded molecules (salt elution, 70–90% of total DNA). 600 µl of 10 mM Tris–HCl pH 8, 1 M NaCl containing 1.8% caffeine were finally added to the column, and after incubation for 10 min, the DNA enriched for ssDNA-containing RIs were eluted from the column. The elution buffer in the caffeine fractions was changed using conical Amicon Ultra centrifugal filters (0.5 ml 100K-MWCO 10K) following manufacturer's instructions. The RIs were resuspended and concentrated in a smaller volume of 10 mM Tris–HCl pH 8. Aliquotes of the samples were then spread onto a water surface in a monomolecular layer in the presence of benzyl-dimethyl-alkyl-ammonium chloride (BAC) and the DNA molecules in the monomolecular layer were adsorbed on carbon-coated EM grids in the presence of uranyl acetate followed by platinum-based low angle rotatory shadowing and analysed as described in ref. [44]. For this dataset a Leica MED020 e-beam evaporator equipped with oil free deep vacuum system (membrane and turbo-molecular pumps), the EVM030 control unit with two EK030 electron beam evaporation sources (Leica catalogue number 16BU007086-T), the QSG100 film thickness monitor (Leica 16LZ03428VN), the

QSK060 Quartz Head (Leica 16LZ03440VN), the Tiltable Rotary Stage (Leica 16BU007283T), and the high precision rotation plate PR01 (ThorLabs Newton, New Jersey, USA) for low angle rotary shadowing were utilized.

The assignment criteria for single-stranded regions on the DNA molecules analysed in this work were described[44]. We note that in order to assign a ssDNA region on a DNA filament it is necessary to identify two points on the DNA molecule that define the borders of the ssDNA region, in which the thickness of the DNA filament (in our experimental conditions ~100 Å, 10 nm) decreases close to one half. We note that the observed thickness of the filaments is largely determined by the amount of deposited heavy atoms during the shadowing procedure. The EM pictures were acquired using a Tecnai 12 BIO TWIN G2 microscope operated at 120 kV with a side mounted GATAN Orius SC-1000 camera. The length measurements were performed using a conversion factor expressed in nanometers/base pair calculated using a plasmid of known size as internal standard[44]. The pixel size was corrected automatically at each magnification according to the internal calibrations of the electron microscope and camera.

**Reporting summary**. Further information on research design is available in the Nature Research Reporting Summary linked to this article.

## Data availability
The authors declare that all data supporting the findings of this study are available within the paper and its supplementary information files. The source data of Figs. 1–7 and Supplementary Figs. 1–7, are provided in the Mendeley datasets (https://doi.org/10.17632/35ksks3k3n.1) and as a Source Data file. The results of the ChIP-on-chip arrays are deposited at the GEO entry (GSE147452), available at https://www.ncbi.nlm.nih.gov/geo/query/acc.cgi?acc=GSE147452. Source data are provided with this paper.

## Code availability
The microarray data have been deposited in the Gene Expression Omnibus (GEO) archive under ID code GSE147452 available at https://www.ncbi.nlm.nih.gov/geo/query/acc.cgi?acc=GSE147452.

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

## Acknowledgements

We thank A. Cocito and the Bioinformatics team at IFOM, F. Ianelli and F. Zanardi, for support with ChIP-on-chip data analysis, D. Piccini for demonstrating the ChIP technique, Cogentech facility for hybridization of Affymetrix chips and production of CEL data files, I. Psakhye and all lab members for discussions. This work was supported by the Italian Association for Cancer Research (IG 23710 and 18976), and European Research Council (Consolidator Grant 682190) grants to D.B. S.A. was partly supported by the AIRC fellowships 20874 and 22255.

## Author contributions

D.B. designed the study. S.A., T.A.C.R., C.R.J., D.M., A.W., and B.S. conducted the experiments. M.G. acquired and analysed DNA structures by EM. S.A., B.S., and D.B. analysed the data and constructed figures. D.B wrote the paper and all authors provided feedback.

## Competing interests

The authors declare no competing interests.
