## [Peer Review File · Nature Communications]

REVIEWER COMMENTS

Reviewer #1 (Remarks to the Author):

Smc5/6 mediates replication completion by coordinating Sgs1-top3-Rmi1 with other resolvases
Agashe, Branzei et al

This manuscript investigates the very complex molecular world of Smc5/6 cohesions, long known or thought to be involved in coordinating difficulties in DNA replication. This manuscript makes some contributions towards knowing about Smc5/6 in budding yeast. I enjoyed the science, though I am less sure that the results really warrant the statement in the title? Other reviewers can help, no doubt. Here is my take, and questions imbedded in italics.

1. This paper seems to have 2 parts. The first part is well-done and clear and relevant to understanding replication biology: components of Smc5/6 chip to the natural pausing sites (NPS), that include centromeres, tRNAs, and termination regions. Also chipping to NPS are components of the STR complex, (Sgs1, Top3, Rmi1) than act to disentangle strands when needed. *In mutants of Smc5/6, the STR complex still binds to NPSs they say...but figure 1e says less Top3 goes to NPS in an smc5-56 mutant?*

2. One feature of these results I found striking was that they performed the chip experiments in G2/M, nocodazole-arrested cells!!! So, the Smc5/6-STR remains at NPSs after DNA replication is done, or rather “done”. *The authors might comment on if they think their chip signal is coming from a few cells with incomplete DNA replication at NPSs, or from many/most cells that have completed replication and have the Smc5/6-STR proteins still at the sites. And, have the authors tried to do such a Chip analysis in synchronized cells, measuring localization in early, middle and late S phase? They do not have to do the experiments...there is already a lot here...but comments, and a figure that makes whatever point they prefer to make.*

3. Still in the first part, the authors then do a second analysis, using HU-delayed cells and examining by 2D DNA gels fork stalling at TER sites. They see joint molecules, etc, and by EM!!!! These are very difficult experiments, so they get full credit for getting them to work.

4. We then moved to the second part, which as far as I can tell is mostly about a mutant in smc6-56(??)—smc6-56 is ts lethal, allowing them to isolate suppressors that grow at 37. They isolate the same missense suppressor several times, and then characterize it (smc6-56sup). They describe its importance-- and need to be a bit more clear in my view to make their key points. It seems the suppressor, which is intragenic seems to nearly completely restore wild type function- it's a hypomorph. The smc6-56sup restores some binding to NPSs, yet restores even greater binding of Top3 to NPS. Then, smc6-56sup cells restores to wild type many functions except for those involves sgs1 and mus81/mms4. Some function is restored to allow sgs1, for example, to live (smc5-56 sgs1 doubles are lethal). So....I am not completely sure what to make of the partial suppression of sgs1 and mus81-mmm4 by smc6-56sup. They need to provide even a hypothetical model, with more than shown in Fig 7, to underscore a major point in this paper that Smc5/6 is coordinating resolvases at NPSs.

Additional Comments.

1. Last sentence in Summary....not “ones” but ‘activity’ ...ones is awkward...☺

2. Page 3..not “exhaustive” is awkward
3. Pg 4 they say Smc5/6 is not required for recruitment of STR....but they show Top3 recruitment is lower in smc56??? I don't get this. I did not see a chip of STR in an smc5-56 mutant in which STR was recruited as in wt??
4. To say that smc5/6 synergizes with Sgs1 to resolve intermediates...I see...more unresolved in smc5-56sup sgs1 than in either single mutant! Aha!! Now I get their argument, based on experiments not doable with smc5-56 cause' its lethal with sgs1.
5. I guess now they can say that Smc5/6 and Mus81-Mms4, and Sgs1 synergize...but they could be acting independently??? And probably are given that Mms4 binds to NPS independent of Smc5/6?

Reviewer #2 (Remarks to the Author):

The role of the Smc5/6 SMC complex has remained enigmatic. Here the authors offer data in support of the model that Smc5/6 works with the STR DNA resolvase and possibly other resolvases in completion of DNA replication at natural pause sites (NPSs). They show that Smc5/6 co-localizes with the STR complex at NPSs, mutations in or depletion of Smc5/6 and STR complexes have similar phenotypes in terms of joint molecule accumulation at stalled replication forks and termination regions. Isolation of an intragenic suppressor of a smc6 mutant allele led to the functional connection between Smc5/6 binding and Top3 binding to chromatin. These studies have led to the model that Smc5/6 regulates DNA resolvases that act at replication termination regions

While these are important conclusions, the presentation of the data and their interpretation is sometimes confusing. Among the issues are the failure to define “resolvases” at the start and to list the known resolvase activities/proteins. Even the title has the word “resolvases” with no adjective to help the reader (such as DNA, crossed strand junction, etc). Next, Top3 has roles independent of Sgs1 but these are not laid out or mentioned until the discussion. Third, the smc6—56-sup allele has both suppressor activity (growth at 37C, how it was isolated), and enhancer activity (Figure 5, DNA damage sensitivities in double mutants), among other figures. This makes it difficult to understand how the suppressor works. Perhaps a table summarizing the smc6-56-sup phenotypes would help.

Other comments:

In the summary the authors speak of an intra-allelic suppressor of smc6-56 but on page 12 they call this an intragenic suppressor. Just call it one thing, please.

Other resolvases. This is a phrase used in the summary and other parts of the text without ever listing them. Indeed, the final figure, Figure 7, shows Yen1 without ever mentioning it specifically or describing it.

Page 3. “Ty structures” are mentioned without defining what Tys are.

Page 3. What is “global” replication?

Page 6. The authors refer to the “cone” signal in Figure 2c but do not indicate this in the cartoon of Figure 2a.

Page 8. Were all the smc6-56-sup isolates independent, as 5/8 had the same additional suppressing mutation?

Page 9. Define “nascent invasions”.

Page 11. What are the ways in which ATR/Mec1 prevent fragility at stalled replication forks? This needs to be understood in order to appreciate how Smc5/6 and STR are different.

Reviewer #3 (Remarks to the Author):

In this manuscript, Agashe et al report a role for Smc5/6 in mediating replication completion by coordinating the activities of the STR complex and the activities of resolvases. The work is generally of high quality and I also find it interesting. I do, however, believe that additional work should be performed to strengthen some of the conclusions and to provide more insight into the precise function(s) of Smc5/6.

Specific comments:

- The authors write that “Sgs1 profiles were not analysed due to poor enrichment of Sgs1 on chromatin.” It is unclear why would this be the case. Which tags did the authors use? What failed? Without knowing if Sgs1 colocalises with Top3-Rmi1 (TR), it is very hard to make some of the conclusions in the manuscript. TR is known to have Sgs1-independent functions and both Top3 and Rmi1 are significantly more abundant than Sgs1. Thus, data for TR cannot be simply extended to STR. In agreement with this, the authors also report Sgs1-independent functions of Top3 (without citing previous work, e.g. Tang et al, 2015; Kaur et al., 2015; etc). Isn't it possible that Sgs1 is not detected at NPSs simply because it isn't there? Couldn't this explain the weaker phenotype of sgs1 mutants compared to top3 mutants?

- In the 2D gels (as well as in other experiments), the authors did not combine Smc5/6 mutations with STR mutations, which is critical to determine whether both complexes function in the same pathway. Conditional mutants are needed, but this should be possible to obtain.

- The authors describe a new allele of Smc6, Smc6-56-sup, which leads to defects in Smc6 retention on chromatin but that restores normal Top3 binding. While this description is accurate, I believe that the interpretation of the data has to be more careful. Smc6-56-sup is clearly able to bind chromatin better than Smc6-56 (>2X better). Thus, it is possible that this is sufficient to fully compensate for the recruitment of Top3, etc, while being insufficient for other functions of Smc6.

- Do the authors know how does the suppression work? How do the overall protein levels of Smc6-56-sup compare to Smc6-WT and Smc6-56? At 37°C?

- One of the main findings by the authors is that Smc5/6 regulates both STR and MUS81. While this is novel in the specific context of NPSs, it is generally/conceptually not entirely novel. Various studies, including Xaver et al 2013, have suggested something very similar during meiotic recombination. In my opinion, it would be important that the authors provide more mechanistic insight into how Smc5/6 coordinates the function of STR and the resolvases.

Minor comments:

- Why do the authors use "pre-anaphase" as a term to describe G2/M cells arrested with NOC?

- The authors refer to a plethora of Smc mutants without explaining what the mutations are. It is extremely hard for the reader to understand such experiments without more information on the specific properties of the mutants. E.g. What does "the constitutive smc6-P4 mutation" mean?

Reviewer #1 (Remarks to the Author):

Smc5/6 mediates replication completion by coordinating Sgs1-top3-Rmi1 with other resolvases
Agashe, Branzei et al

This manuscript investigates the very complex molecular world of Smc5/6 cohesions, long known or thought to be involved in coordinating difficulties in DNA replication. This manuscript makes some contributions towards knowing about Smc5/6 in budding yeast. I enjoyed the science, though I am less sure that the results really warrant the statement in the title? Other reviewers can help, no doubt. Here is my take, and questions imbedded in italics.

We are happy that the reviewer enjoyed our science. It is true that the paper has a strong focus on Smc5/6 and STR and therefore we are changing now the title to “Smc5/6 functions with the Sgs1-Top3-Rmi1 DNA resolvase to complete chromosome replication at natural pause sites”

1. This paper seems to have 2 parts. The first part is well-done and clear and relevant to understanding replication biology: components of Smc5/6 chip to the natural pausing sites (NPS), that include centromeres, tRNAs, and termination regions. Also chipping to NPS are components of the STR complex, (Sgs1, Top3, Rmi1) than act to disentangle strands when needed. In mutants of Smc5/6, the STR complex still binds to NPSs they say...but figure 1e says less Top3 goes to NPS in an smc5-56 mutant?

We thank the reviewer for the comment. Indeed, Top3 can still localize to NPSs genome-wide in Smc5/6 mutants as addressed by ChIP-on-chip analysis (Supplementary Fig. 1d), but the amount of Top3 binding is decreased as observed by quantitative ChIP-qPCR at NPSs (Figure 1e) and the genome-wide coverage (added analysis to Supplementary Fig. 1d). Thus, Smc5/6 facilitates Top3 binding/retention, but it is not essential for its recruitment at NPSs.

2. One feature of these results I found striking was that they performed the chip experiments in G2/M, nocodazole-arrested cells!!! So, the Smc5/6-STR remains at NPSs after DNA replication is done, or rather “done”. The authors might comment on if they think their chip signal is coming from a few cells with incomplete DNA replication at NPSs, or from many/most cells that have completed replication and have the Smc5/6-STR proteins still at the sites. And, have the authors tried to do such a Chip analysis in synchronized cells, measuring localization in early, middle and late S phase? They do not have to do the experiments...there is already a lot here...but comments, and a figure that makes whatever point they prefer to make.

In nocodazole-arrested cells, cells reach G2/M and have completed bulk replication. As NPSs are being replicated late, we envisage that Smc5/6-STR is being present at these sites to finalize their replication and mediate replication termination. This process may be similar with what happens at replication forks stalled by high doses of HU. We performed ChIP-qPCR of Smc6 and Top3 at regions replicating under HU conditions, when the replication forks stall in close proximity to early origins of replication from which they originate, and find that Smc5/6-STR are similarly enriched at those sites (Supplementary Figs. 2b and 2c).

3. Still in the first part, the authors then do a second analysis, using HU-delayed cells and examining by 2D DNA gels fork stalling at TER sites. They see joint molecules, etc, and by EM!!!! These are very difficult experiments, so they get full credit for getting them to work.

We are happy that the reviewer positively evaluated this difficult task.

4. We then moved to the second part, which as far as I can tell is mostly about a mutant in *smc6-56*(??)—*smc6-56* is ts lethal, allowing them to isolate suppressors that grow at 37. They isolate the same missense suppressor several times, and then characterize it (*smc6-56sup*). They describe its importance-- and need to be a bit more clear in my view to make their key points. It seems the suppressor, which is intragenic seems to nearly completely restore wild type function- it's a hypomorph. The *smc6-56sup* restores some binding to NPSs, yet restores even greater binding of Top3 to NPS. Then, *smc6-56sup* cells restores to wild type many functions except for those involves *sgs1* and *mus81/mms4*. Some function is restored to allow *sgs1*, for example, to live (*smc5-56 sgs1* doubles are lethal). So....I am not completely sure what to make of the partial suppression of *sgs1* and *mus81-mms4* by *smc6-56sup*. They need to provide even a hypothetical model, with more than shown in Fig 7, to underscore a major point in this paper that *Smc5/6* is coordinating resolvases at NPSs.

We revisited the *smc6-56-sup* part to present it in a clearer fashion and performed new experiments to improve our understanding of this mutant and what it reveals about *Smc5/6*. One of the compelling findings of this part is that together with a good recovery in *Smc5/6* function in the *smc6-56-sup* suppressor, we alleviate Top3 binding (as observed by ChIP-qPCR at NPSs and now extended also to genome-wide clusters, see Fig. 4d), strengthening the notion of a tight connection between *Smc5/6* and Top3. We agree with the reviewer that *smc6-56-sup* is a hypomorph, with many functions that are defective in *smc6-56* being recovered to good levels in the suppressor allele. The synthetic lethality between *smc6-56* and *sgs1*, *mus81* is relieved with *smc6-56-sup*, although some additivity is still observed in the presence of HU due to partial loss of function in *smc6-56-sup*. Notably, however, besides being a hypomorph, *smc6-56-sup* becomes defective in a function that prevents, complementarily with *Sgs1* and Top3, JM accumulation at NPSs. To reach this conclusion, we performed 2D gels with *sgs1Δ*, conditional *sgs1* and *top3* alleles combined with *smc6-56-sup* (Fig. 5c, 6b, Supplementary Fig. 6d) or *smc6-56* (Supplementary Fig. 4). The enhancer phenotype in terms of persisting JMs is observed with *smc6-56-sup*, although we note that *smc6-56* could only be analyzed in combination with conditional *Tc-sgs1* due to synthetic lethality with *sgs1Δ*. Thus, we infer that *smc6-56-sup* becomes defective in pathway(s) that counteract JM accumulation at NPSs. Genetically, we find that the defects of *smc6-56-sup* do not synergize with loss of *Srs2* and *Mph1*, mutations in which have negative genetic interactions with *Sgs1* and *Mus81-Mms4* impairment. Thus, *smc6-56-sup* defects may be related to *Srs2* and *Mph1* actions. The persistence in JMs implies also a defect in resolving JMs by independent pathways, such as via *Srs2* or by the *Mus81-Mms4* nuclease, which we find to colocalize a NPSs (Fig. 7c-e). Single mutations in *Srs2* and *Mms4* do not cause the same phenotype with *smc6-56-sup* in terms of JM accumulation at NPSs (Supplementary Fig. 7b), indicating that the defect in *smc6-56-sup* is likely compounded. We propose that *Smc5/6* may coordinate early joint molecule disruption activities (defined by *Srs2*) and inhibition of fork remodeling (*Mph1*) with double Holliday junction dissolvase (Top3 and STR) and Holliday junction resolvase (*Mus81-Mms4*) activities to facilitate replication termination at NPSs, as discussed in the new model we are proposing (Fig. 8).

Additional Comments.

1. Last sentence in Summary....not “ones” but ‘activity’...ones is awkward...☒

Modified

2. Page 3..not “exhaustive” is awkward

Removed

3. Pg 4 they say *Smc5/6* is not required for recruitment of STR....but they show Top3 recruitment is lower in *smc56*??? I don't get this. I did not see a chip of STR in an *smc5-56* mutant in which STR was recruited as in wt??

As discussed in the point-by-point response, Top3 binding is quantitatively reduced in *smc5/6* mutants but there is still statistically significant binding at NPSs genome-wide.

4. To say that *smc5/6* synergizes with *Sgs1* to resolve intermediates...I see...more

unresolved in *smc5-56sup sgs1* than in either single mutant! Aha!! Now I get their argument, based on experiments not doable with *smc5-56* cause' its lethal with *sgs1*.

Yes, indeed, we discussed this above, along with the new results obtained by combining *smc6-56* with conditional *sgs1*.

5. I guess now they can say that Smc5/6 and Mus81-Mms4, and Sgs1 synergize...but they could be acting independently??? And probably are given that Mms4 binds to NPS independent of Smc5/6?

We showed initially that Smc5/6 does not affect Mms4 binding at NPSs as we can measure by ChIP-qPCR (Fig. 7f), which would argue that the binding of Mms4 is largely independent of Smc5/6. Based on increased persistence of joint molecules in *smc6-56-sup sgs1/top3* mutants compared to *sgs1/top3* mutants, we reason that other activities are impaired in this Smc5/6 mutant, possibly Srs2 and Mus81. It is most probable, as we are suggesting in the new proposed model and in the answer above, that Smc5/6 needs to coordinate D-loop disruption activities with resolution activities, including the one of STR, to facilitate replication termination.

Reviewer #2 (Remarks to the Author):

The role of the Smc5/6 SMC complex has remained enigmatic. Here the authors offer data in support of the model that Smc5/6 works with the STR DNA resolvase and possibly other resolvases in completion of DNA replication at natural pause sites (NPSs). They show that Smc5/6 co-localizes with the STR complex at NPSs, mutations in or depletion of Smc5/6 and STR complexes have similar phenotypes in terms of joint molecule accumulation at stalled replication forks and termination regions. Isolation of an intragenic suppressor of a *smc6* mutant allele led to the functional connection between Smc5/6 binding and Top3 binding to chromatin. These studies have led to the model that Smc5/6 regulates DNA resolvases that act at replication termination regions

While these are important conclusions, the presentation of the data and their interpretation is sometimes confusing. Among the issues are the failure to define “resolvases” at the start and to list the known resolvase activities/proteins. Even the title has the word “resolvases” with no adjective to help the reader (such as DNA, crossed strand junction, etc). Next, Top3 has roles independent of Sgs1 but these are not laid out or mentioned until the discussion. Third, the *smc6-56-sup* allele has both suppressor activity (growth at 37C, how it was isolated), and enhancer activity (Figure 5, DNA damage sensitivities in double mutants), among other figures. This makes it difficult to understand how the suppressor works. Perhaps a table summarizing the *smc6-56-sup* phenotypes would help. We are happy that the reviewer liked our work and considers our findings important. We paid attention to define DNA resolvases and describe the ones analyzed here, point similarities and differences between Sgs1 and Top3 and to clarify the phenotypes of *smc6-56-sup*. The last task involved substantial experimentation. In regard to how the suppressor works, we interpret our results to indicate that *smc6-56-sup* is a hypomorph, with many functions defective in *smc6-56* recovered to nearly wt levels in the suppressor allele. The synthetic lethality between *smc6-56* and *sgs1*, *mus81* is relieved with *smc6-56-sup*, although some additivity is still observed in the presence of HU due to partial loss of function in *smc6-56-sup*. Notably, however, besides being a hypomorph, *smc6-56-sup* becomes newly or more defective in a function that prevents, complementarily with Sgs1 and Top3, JM accumulation at NPSs. To reach this conclusion, we performed 2D gels with conditional *sgs1* allele combined with *smc6-56* (Supplementary Fig. 4) or *sgs1Δ*, conditional *sgs1* and *top3* alleles combined with *smc6-56-sup* (Fig. 5c, 6b, Supplementary Fig. 6d). The enhancer phenotype in terms of persisting JMs is observed specifically with *smc6-56-sup*. Thus, we infer that *smc6-56-sup* becomes defective in pathway(s) that counteract JM accumulation at NPSs in manners that backup STR. Genetically, we find that the defects of *smc6-56-sup* do not synergize with loss of Srs2 and Mph1, mutations in which have negative genetic interactions with Sgs1 and Mus81-Mms4 impairment. Thus, *smc6-56-sup* defect(s) may be related to Srs2 and Mph1 actions. The persistence in JMs implies also a defect in resolving JMs by independent pathways, such as via Srs2 or by the Mus81-Mms4 nuclease, which we find to colocalize a NPSs (Fig.7c-e). Single mutations in Srs2 and Mms4 do not cause the same phenotype with *smc6-56-sup* in terms of JM accumulation at NPSs (Supplementary Fig. 7b), indicating that the defect in *smc6-56-sup* is likely compounded. We propose that Smc5/6 may coordinate early joint molecule disruption activities (defined by Srs2) and inhibition of fork remodeling (Mph1) with double Holliday junction dissolvase (Top3 and STR) and Holliday junction resolvase (Mus81-Mms4) activities to facilitate replication termination at NPSs, as discussed in the new model we are proposing (Fig. 8).

Other comments:

In the summary the authors speak of an intra-allelic suppressor of *smc6-56* but on page 12 they call this an intragenic suppressor. Just call it one thing, please. We are referring now to it as an intra-allelic suppressor throughout the manuscript.

Other resolvases. This is a phrase used in the summary and other parts of the text without ever listing them. Indeed, the final figure, Figure 7, shows Yen1 without ever mentioning it specifically or describing it. We are now describing the known resolvase activities/proteins and do not comment further on Yen1 that was not specifically analyzed in this study.

Page 3. “Ty structures” are mentioned without defining what Tys are. We are defining Tys.

Page 3. What is “global” replication? We changed it for “bulk replication”.

Page 6. The authors refer to the “cone” signal in Figure 2c but do not indicate this in the cartoon of Figure 2a. We have added the cone signal to the cartoon in Figure 2a.

Page 8. Were all the *smc6-56-sup* isolates independent, as 5/8 had the same additional suppressing mutation? Yes, they come from 5 independently isolated clones that had the same suppressor mutation.

Page 9. Define “nascent invasions”. We rewrote this part.

Page 11. What are the ways in which ATR/Mec1 prevent fragility at stalled replication forks? This needs to be understood in order to appreciate how Smc5/6 and STR are different. We are commenting on previous publications showing evidence for unusual replication fork structures in ATR/Mec1 mutants with hemi-replicated bubbles and resected forks among other intermediates after incubation with high doses of HU for 60 min. Differently from *mec1* and *rad53* mutants, *smc5/6* and *sgs1*, *top3* mutants can tolerate HU incubation for long times, progress into G2/M, but accumulate several types of joint molecules. This analysis indicates that the major defects of STR and Smc5/6 mutants lie in preventing formation and in resolving different types of DNA joint molecules rather than in protecting stalled replication forks from resection and collapse. We have revisited and rewrote this part to improve clarity.

Reviewer #3 (Remarks to the Author):

In this manuscript, Agashe et al report a role for Smc5/6 in mediating replication completion by coordinating the activities of the STR complex and the activities of resolvases. The work is generally of high quality and I also find it interesting. I do, however, believe that additional work should be performed to strengthen some of the conclusions and to provide more insight into the precise function(s) of Smc5/6.

We are happy that the reviewer found our work of high quality and interesting. We are now providing additional data to probe similarities and differences between Sgs1 and Top3, as pointed out by the reviewer in the specific comments, and to probe the functions defective in the isolated *smc6-56* suppressor as means to provide insight into Smc5/6 functions. We also changed the title to focus on the functional interaction between Smc5/6 and STR in completing chromosome replication at NPSs.

Specific comments:

- The authors write that “Sgs1 profiles were not analysed due to poor enrichment of Sgs1 on chromatin.” It is unclear why would this be the case. Which tags did the authors use? What failed? Without knowing if Sgs1 colocalises with Top3-Rmi1 (TR), it is very hard to make some of the conclusions in the manuscript. TR is known to have Sgs1-independent functions and both Top3 and Rmi1 are significantly more abundant than Sgs1. Thus, data for TR cannot be simply extended to STR. In agreement with this, the authors also report Sgs1-independent functions of Top3 (without citing previous work, e.g. Tang et al, 2015; Kaur et al., 2015; etc). Isn't it possible that Sgs1 is not detected at NPSs simply because it isn't there? Couldn't this explain the weaker phenotype of *sgs1* mutants compared to *top3* mutants?

We have previously used 3Flag, 6Flag and 13Myc C-terminal tags for Sgs1 ChIP with poor chromatin enrichment. We reasoned that helicases such as Sgs1 may not need to be stably associated with DNA. In addition, C-terminal tagging of Sgs1 may impair its function as reported in (Cohen and Lichten, G3, 2020). During the revision we used an N-terminal 6HA tag, which is also linked to a stronger *ADHI* promoter, with which we performed ChIP-on-chip with good confidence to add the results in the Supplementary Fig. 1a and align them to those of Top3 and Rmi1. We observe statistically significant genome-wide overlap in Sgs1 clusters with those of Top3 and Rmi1. Moreover, 58 out of 71 TERs contains Sgs1, similarly to Smc6, but do not have statistically significant overlap with tRNAs (Supplementary Fig. 1a, b, c). We are now discussing also the papers describing Sgs1-independent roles of Top3 (Tang et al, 2015; Kaur et al., 2015; Fasching et al, 2015). We like to note that the only setting in which *top3* mutants have a stronger defect compared to *sgs1* is related to the hemicatenane accumulation, compensated by higher levels of dHJs and reversed forks in *sgs1* mutants (Figure 3). Overall, the genome-wide binding patterns of Sgs1 and Top3 and the 2D gel and EM phenotypes of the mutants are very similar. Therefore, while we reveal that Smc5/6 has strong ties with Top3, many of Top3 roles related to replication through NPSs are likely performed in the context of the Sgs1-Top3-Rmi1 complex.

- In the 2D gels (as well as in other experiments), the authors did not combine Smc5/6 mutations with STR mutations, which is critical to determine whether both complexes function in the same pathway. Conditional mutants are needed, but this should be possible to obtain.

We have now made and analyzed these mutants. We observe that *smc6-56* and *sgs1* conditional mutants behave very similarly to the single mutants (see Supplementary Fig. 4). The notion of joint action between STR and Smc5/6 is supported also by the EM analysis of the types and frequencies on intermediates upon depletion of STR or Smc5/6 and by other intricate analysis made possible

with the new *smc6-56* suppressor, which suppresses several *smc6-56* phenotypes, including the reduced Top3 binding.

- The authors describe a new allele of Smc6, *Smc6-56-sup*, which leads to defects in Smc6 retention on chromatin but that restores normal Top3 binding. While this description is accurate, I believe that the interpretation of the data has to be more careful. *Smc6-56-sup* is clearly able to bind chromatin better than *Smc6-56* (>2X better). Thus, it is possible that this is sufficient to fully compensate for the recruitment of Top3, etc, while being insufficient for other functions of Smc6.

We thank the reviewer for this comment. We agree largely with this interpretation, now further strengthened by ChIP-on-chip results of Top3 in wt, *smc6-56* and *smc6-56-sup* (Fig. 4d). Moreover, we combined *smc6-56-sup* (devoid of a specific 2D gel phenotype on its own) not only with *sgs1Δ* and *Tc-sgs1* (results present in the original manuscript), but also with *Tc-top3-AID* and find in all cases a more severe accumulation of joint molecules (Fig. 5c, 6b, Supplementary Fig. 6d). These results point out that *smc6-56-sup* becomes defective in a pathway that acts complementarily with both Sgs1 and Top3 in preventing accumulation of joint molecules and which rationally needs to be defined by other DNA resolvases. Thus, we think that along Top3 functional restoration, *smc6-56-sup* becomes defective in other pathways that act complementarily with Sgs1 and Top3 to prevent joint molecule accumulation at NPSs.

- Do the authors know how does the suppression work? How do the overall protein levels of *Smc6-56-sup* compare to *Smc6-WT* and *Smc6-56*? At 37°C?

We have now performed these experiments and find partial suppression of the *Smc6-56* protein stability at 37°C in *Smc6-56-sup*, a result we added in Supplementary Fig. 5c.

- One of the main findings by the authors is that *Smc5/6* regulates both STR and MUS81. While this is novel in the specific context of NPSs, it is generally/conceptually not entirely novel. Various studies, including Xaver et al 2013, have suggested something very similar during meiotic recombination. In my opinion, it would be important that the authors provide more mechanistic insight into how *Smc5/6* coordinates the function of STR and the resolvases.

Indeed, we have cited the study of Xaver et al, 2013, which proposes a role for *Smc5/6* in regulating Mus81 activity in meiosis. Here, with the help of the *smc6-56-sup* allele, we found that *Smc5/6* regulates the levels of joint molecules arising at NPSs regions in mitosis in a manner complementary with the one provided by the STR complex. We proposed that Mus81 may be involved based on several studies highlighting its role in mitosis to remove joint molecules and on our finding that it localizes to NPSs and TER regions, which however is independent of *Smc5/6* functionality (Fig. 7c-f). Moving forward, we further assessed if removal of Mms4-Mus81 phenocopies the *smc6-56-sup* allele, but this was not the case (Supplementary Fig. 7b). Moreover, using genetics, we reveal that *smc6-56-sup* aggravates the HU sensitivity of both *sgs1* and *mms4/mus81* mutants (Fig. 5a-b), but not those of *srs2* and *mph1* (Fig. 7a-b), which have negative genetic interactions with Sgs1 and Mus81-Mms4 mutants. Thus, the defects of *smc6-56-sup* may be related to those of Srs2 and Mph1. In the context of *sgs1* and *top3* mutants, this should cause more joint molecules to be resolved by STR and other resolvases. We find that single *srs2* and *mms4* mutants do not cause the phenotype of *smc6-56-sup* in terms of joint molecule accumulation at NPSs in *Tc-sgs1* mutants (Supplementary Fig. 7b), suggesting that the defect of *smc6-56-sup* is compounded. We propose that *Smc5/6* may coordinate early joint molecule disruption activities (defined by Srs2) and fork reversal (by Mph1) with Holliday junction resolvase and dHJ dissolvase activities that are present at termination regions and NPSs, as discussed in a new model we are proposing (Fig. 8).

Minor comments:

- Why do the authors use “pre-anaphase” as a term to describe G2/M cells arrested with NOC?

We are now using the term G2/M.

- The authors refer to a plethora of Smc mutants without explaining what the mutations are. It is extremely hard for the reader to understand such experiments without more information on the specific properties of the mutants. E.g. What does “the constitutive *smc6*-P4 mutation” mean?

We are explaining now the information available on the *smc6* mutants. The adjective “constitutive” was used to contrast the nature of this mutation with other conditional mutants. As the reviewer pointed out, it is not necessary and we removed the word “constitutive”.

REVIEWERS' COMMENTS

Reviewer #1 (Remarks to the Author):

The authors addressed my issues.

I really do not understand Fig 8 as much as I would like to. Fig 8D...Smc5/6 are activating Srs2 and STR...and they do what? And c to d...there is some fork reversal, then reinvasion? And, Fig 8d also has the leading strand on the bottom duplex with this odd kink in it, which as far as I can tell is not needed?

Reviewer #2 (Remarks to the Author):

The authors have performed additional experiments and rewritten sections of the manuscript. It is now better presented and makes an interesting story on Smc5/6 with STR at NPS. The findings should be applicable to mammalian systems. The analysis of the smc5-56 intra-allelic suppressor seems under-developed but additional work to expand on this is beyond the scope of this manuscript. Presumably some biochemistry will be done on the mutant protein at some point. Just as a brief note, the figure legend title to S6 should be changed as Srs2 and Mph1 are helicases (stated on page 12), not resolvases.

Reviewer #3 (Remarks to the Author):

The authors have satisfactorily addressed the reviewers comments and have improved the manuscript substantially.

Minor comments:

1) In the title and abstract the authors use "DNA crossed-strand resolvase Sgs1-Top3-Rmi1 (STR)". I understand what the authors mean, but the term resolvase is generally reserved for nucleases. I would suggest to avoid the term resolvase when referring to STR.

2) In the abstract the authors write " Thus, Smc5/6 functions jointly with Top3 and STR to mediate replication completion and influences the activity of other DNA resolvases at NPSs". This is just a detail, but I would not use the term "activity" here. I would replace it by "function".

Response to reviewers' comments for NCOMMS-20-35511-A

Reviewer #1 (Remarks to the Author):

The authors addressed my issues.

I really do not understand Fig 8 as much as I would like to. Fig 8D...Smc5/6 are activating Srs2 and STR...and they do what? And c to d...there is some fork reversal, then reinvasion? And, Fig 8d also has the leading strand on the bottom duplex with this odd kink in it, which as far as I can tell is not needed?

We thank the reviewer for the comments. We wrote in the figure legends the activities enhanced or prevented by Smc5/6, we are now writing a simple explanation also in Figures 8c and 8d to facilitate the immediate understanding of the proposed model and the activities we hypothesize to be modulated by Smc5/6. Moreover, we are adding more explanatory information in the Figure legend about the dashed lines that represent repetitive regions and the potential toxic strand invasion within the non-replicated region thought to be triggered by an initial fork reversal. The kink in Figure 8d is to illustrate that such invasion may have been triggered by reversed fork formation (potentially mediated by Mph1, as illustrated in panel 8c), as now more elaborated in the Figure legend.

Reviewer #2 (Remarks to the Author):

The authors have performed additional experiments and rewritten sections of the manuscript. It is now better presented and makes an interesting story on Smc5/6 with STR at NPS. The findings should be applicable to mammalian systems. The analysis of the smc5-56 intra-allelic suppressor seems under-developed but additional work to expand on this is beyond the scope of this manuscript. Presumably some biochemistry will be done on the mutant protein at some point. Just as a brief note, the figure legend title to S6 should be changed as Srs2 and Mph1 are helicases (stated on page 12), not resolvases.

We thank the reviewer for the supportive comments. We changed the figure legend of Supplementary Fig. 6.

Reviewer #3 (Remarks to the Author):

The authors have satisfactorily addressed the reviewers comments and have improved the manuscript substantially.

Minor comments:

1) In the title and abstract the authors use "DNA crossed-strand resolvase Sgs1-Top3-Rmi1 (STR)". I understand what the authors mean, but the term resolvase is generally reserved for nucleases. I would suggest to avoid the term resolvase when referring to STR.

2) In the abstract the authors write "Thus, Smc5/6 functions jointly with Top3 and STR to mediate replication completion and influences the activity of other DNA resolvases at NPSs". This is just a detail, but I would not use the term "activity" here. I would replace it by "function".

We thank the reviewer for the supportive comments. We changed the terminology both in the title and abstract.